# Cell-type-specific control of basolateral amygdala neuronal circuits via entorhinal cortex-driven feedforward inhibition

E Mae Guthman[1,2], Joshua D Garcia[3], Ming Ma[4], Philip Chu[2,5], Serapio M Baca[2,6], Katharine R Smith[3], Diego Restrepo[1†], Molly M Huntsman[1,2,7†]*

[1]Neuroscience Graduate Program, University of Colorado Anschutz Medical Campus, Aurora, United States; [2]Department of Pharmaceutical Sciences, University of Colorado Anschutz Medical Campus, Aurora, United States; [3]Department of Pharmacology, University of Colorado Anschutz Medical Campus, Aurora, United States; [4]Department of Cell and Developmental Biology, University of Colorado Anschutz Medical Campus, Aurora, United States; [5]Department of Neurosurgery, University of Colorado Anschutz Medical Campus, Aurora, United States; [6]Department of Neurology, University of Colorado Anschutz Medical Campus, Aurora, United States; [7]Department of Pediatrics, University of Colorado Anschutz Medical Campus, Aurora, United States

*For correspondence:
molly.huntsman@CUAnschutz.edu

†These authors contributed equally to this work

Competing interests: The authors declare that no competing interests exist.

**Abstract** The basolateral amygdala (BLA) plays a vital role in associating sensory stimuli with salient valence information. Excitatory principal neurons (PNs) undergo plastic changes to encode this association; however, local BLA inhibitory interneurons (INs) gate PN plasticity via feedforward inhibition (FFI). Despite literature implicating parvalbumin expressing (PV[+]) INs in FFI in cortex and hippocampus, prior anatomical experiments in BLA implicate somatostatin expressing (Sst[+]) INs. The lateral entorhinal cortex (LEC) projects to BLA where it drives FFI. In the present study, we explored the role of interneurons in this circuit. Using mice, we combined patch clamp electrophysiology, chemogenetics, unsupervised cluster analysis, and predictive modeling and found that a previously unreported subpopulation of fast-spiking Sst[+] INs mediate LEC→BLA FFI.

## Introduction

The ability of animals to learn to associate sensory stimuli with outcomes plays an important role in their survival. Specifically, animals learn to associate environmental stimuli with particular valences (rewarding or aversive outcomes). The BLA is key in the formation of these associations (*Duvarci and Pare, 2014*; *Janak and Tye, 2015*). Following a small number of presentations of a stimulus and valence information, such as a reward or threatening foot-shock, BLA excitatory PNs undergo plastic changes. For example, after a few trials BLA neurons fire selectively to novel odors that are informative about the outcome in an associative olfactory learning task (*Schoenbaum et al., 1999*). This plasticity is critically important as future presentations of the stimulus can then elicit robust firing in subsets of BLA PNs that output to downstream regions to guide goal-directed behavior (*Beyeler et al., 2016*; *Janak and Tye, 2015*; *Quirk et al., 1995*).

Accumulating evidence demonstrates that this learning induced-plasticity is input-specific (*Kim and Cho, 2017*; *McKernan and Shinnick-Gallagher, 1997*; *Nabavi et al., 2014*); however, the circuit mechanisms underlying this input specific plasticity remain unknown. Importantly, GABAergic inhibition regulates the ability of BLA PNs to undergo LTP with plasticity unable to occur in the presence of intact inhibition (*Bissière et al., 2003*; *Krabbe et al., 2018*). Further, local dendritic

targeting GABAergic INs expressing Sst exert strong control over BLA learning processes in vivo: Sst[+] INs are normally inhibited for the duration of the sensory stimulus-valence pairing and their activation impairs learning (*Wolff et al., 2014*). In acute slice preparations, FFI exerts this inhibitory control over BLA plasticity (*Bazelot et al., 2015*; *Bissière et al., 2003*; *Tully et al., 2007*). In cerebral cortex and hippocampus, FFI is mediated by perisomatic targeting, PV[+] INs to ensure the temporal fidelity of PN action potential (AP) output (*Glickfeld and Scanziani, 2006*; *Pouille and Scanziani, 2001*; *Tremblay et al., 2016*). However, in BLA, FFI plays a dual role regulating PN AP output (*Lang and Paré, 1997*) and gating their plasticity (*Bazelot et al., 2015*; *Bissière et al., 2003*; *Tully et al., 2007*). Which IN population mediates FFI to control plasticity in BLA remains unclear: perisomatic targeting PV[+] INs are candidates as they mediate FFI in cerebral cortex and hippocampus (*Glickfeld and Scanziani, 2006*; *Pouille and Scanziani, 2001*; *Tremblay et al., 2016*), but anatomical studies suggest that Sst[+] INs may play this role in the BLA (*Smith et al., 2000*; *Unal et al., 2014*). Further, the role of Sst[+] INs in regulating BLA learning in vivo (*Wolff et al., 2014*) suggests a role in regulating BLA plasticity via FFI.

The LEC receives extensive innervation from the olfactory bulb (*Igarashi et al., 2012*) and piriform cortex (*Johnson et al., 2000*), and neurons in this area respond to odorants (*Leitner et al., 2016*; *Xu and Wilson, 2012*). Restricted firing in LEC suggests that it may play a role in modulating odor-specific, experience- and state-dependent olfactory coding (*Xu and Wilson, 2012*). LEC stimulation exerts an inhibitory effect on the olfactory input from the olfactory bulb to BLA and, to a lesser extent, piriform cortex (*Mouly and Di Scala, 2006*). Given that BLA is involved in encoding the motivational significance of olfactory cues in associative learning (*Schoenbaum et al., 1999*) and that the LEC→BLA synapse is plastic (*Yaniv et al., 2003*), the LEC→BLA circuit could be involved in plasticity during olfactory learning. In addition to its role in olfactory processing, the LEC plays an important role in multimodal sensory processing (*Keene et al., 2016*; *Tsao et al., 2013*) and the BLA is a major target of LEC efferents (*McDonald, 1998*). Finally, the LEC projects along the perforant pathway to engage the hippocampal trisynaptic circuit and receives input to its deep layers from CA1 (*Neves et al., 2008*). In turn, it is these deep layer LEC neurons that innervate the BLA (*McDonald, 1998*; *McDonald and Mascagni, 1997*). Taken together, this raises the possibility that the LEC→BLA circuit could be involved in sensory-valence learning across sensory modalities or for more multimodal natural stimulus information under hippocampal influence. Here we address the circuit mechanisms underlying BLA dependent learning by studying the role of distinct populations of inhibitory interneurons in mediating LEC→BLA circuit motifs.

Due to the relative scarcity of cortical afferent synapses onto BLA PV[+] INs (*Smith et al., 2000*), the role of BLA Sst[+] INs in regulating learning in vivo (*Wolff et al., 2014*), and the importance of FFI circuits in gating BLA LTP (*Bazelot et al., 2015*; *Bissière et al., 2003*; *Tully et al., 2007*), we hypothesized that Sst[+] INs provide FFI onto local BLA PNs. We tested this hypothesis by pairing patch clamp recordings in BLA with unsupervised cluster analysis, predictive modeling, and chemogenetic manipulations and found that a previously unreported subpopulation of fast spiking (FS) Sst[+] INs mediate FFI in the LEC→BLA circuit.

## Results

### LEC afferents preferentially drive disynaptic FFI in BLA

To examine how LEC afferents, a major cortical source of polysynaptic inhibition to BLA (*Lang and Paré, 1997*), engaged BLA neuronal circuitry, we prepared acute horizontal slices of mouse brain containing both BLA and LEC and recorded synaptic responses of BLA PNs (*Figure 1A, B*). A single stimulation of LEC elicited evoked EPSCs (EPSCs) and IPSCs (IPSCs) in PNs (*Figure 1B*). EPSCs were blocked by glutamate receptor antagonists DNQX (20 μM) and D-APV (50 μM) but not by the GABA_A receptor antagonist gabazine (gbz; 5 μM) (*Figure 1C, D*; $p = 6.72 \times 10^{-4}$, Kruskal-Wallis test). In contrast, IPSCs were blocked by either gbz or DNQX/APV (*Figure 1C, D*; $p = 6.45 \times 10^{-4}$, Kruskal-Wallis test), consistent with a monosynaptic glutamatergic nature of the EPSCs and a polysynaptic GABAergic nature of the IPSCs. Additionally, IPSC onset was delayed relative to EPSC onset by ~3 ms (*Figure 1E*; $p = 0.023$, paired t-test) consistent with prior reports of polysynaptic inhibitory circuits in BLA (*Arruda-Carvalho and Clem, 2014*; *Hübner et al., 2014*; *Lucas et al., 2016*).

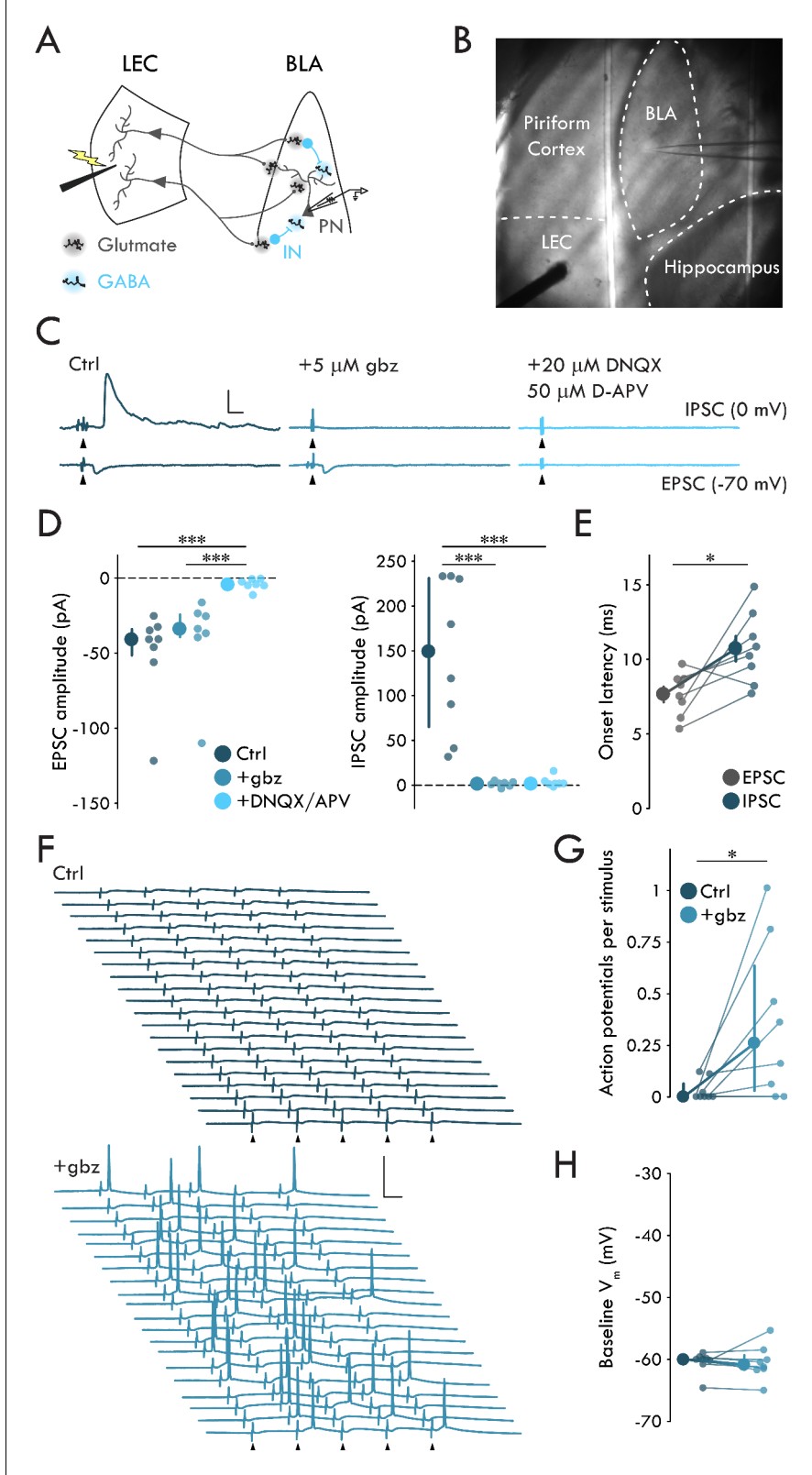

**Figure 1.** LEC afferents preferentially drive disynaptic FFI in BLA. (**A**) Experimental schematic. (**B**) Representative image of experimental preparation. LEC and BLA as well as neighboring hippocampus and piriform cortex are labeled. The external capsule can be seen running laterally of BLA and forming its border with piriform cortex. Note the stimulation electrode positioned in LEC and recording pipette in BLA. (**C**) Representative traces of
*Figure 1 continued on next page*

*Figure 1 continued*

EPSCs and IPSCs in a BLA PN in response to LEC stimulation (top: control; middle: gbz, 5 μM; bottom: DNQX, 20 μM, D-APV, 50 μM). Arrowheads: stimulation (artifacts truncated). Scale bars: 100 pA, 10 ms. (**D**) EPSCs blocked by DNQX/APV (Kruskal-Wallis test: $p = 6.72 \times 10^{-4}$; $n_{control}$ = 8, 4, $n_{gbz}$ = 7, 4, $n_{DNQX/APV}$ = 7, 4). IPSCs blocked by gbz and DNQX/APV (Kruskal-Wallis test: $p = 6.45 \times 10^{-4}$; $n_{control}$ = 8, 4, $n_{gbz}$ = 7, 4, $n_{DNQX/APV}$ = 7, 4). (**E**) Onset latency of IPSC is delayed relative to the EPSC (paired t-test: p=0.023, n = 8, 4). (**F**) 20 voltage traces from a representative BLA PN (top: control; bottom: gbz) in response to 5 × stimulation of LEC at 20 Hz. Arrowheads: stimulation. Scale bars: 40 mV, 20 ms. (**G**) $GABA_A$ receptor blockade increases AP firing in BLA PNs in response to LEC stimulation (Wilcoxon signed-rank test: p=0.031, n = 8, 3). (**H**) Baseline $V_m$ did not differ between control and gbz (Wilcoxon signed-rank test: p=0.38, n = 8, 3). Summary statistics in E presented as mean ± standard error of the mean (s.e.m.). Summary statistics in D, G, H presented as median with interquartile range (IQR). Individual data points presented adjacent to the summary statistics. *p < 0.05, ***p < 0.001. See *Figure 1—source data 1* for a table with full details on all statistical tests used in this figure. See *Figure 1—source data 2* for a table of all individual data points displayed in *Figure 1*.

The online version of this article includes the following source data for figure 1:

**Source data 1.** Table of statistical analyses used in *Figure 1*.
**Source data 2.** Table of data included in *Figure 1*.

The two main archetypal circuit motifs that mediate polysynaptic inhibition are feedforward and feedback inhibition (*Tremblay et al., 2016*). In order to distinguish these motifs, we recorded from BLA PNs in response to a stimulus train of five pulses to the LEC at 20 Hz. If eIPSCs were the result of FFI alone, we would expect to see little to no AP activity in BLA PNs in response to stimulation. However, if the eIPSCs were a result of combined feedforward and feedback inhibition, we would expect to see the PNs fire in response to LEC stimulation. Recording from BLA PNs in current clamp (−60 mV), we found they rarely fired in response to stimulation (*Figure 1F,G*). These data suggest LEC stimulation preferentially drives FFI in BLA PNs.

To determine whether our experimental setup recapitulates LEC suppression of BLA PNs, as found in vivo (*Lang and Paré, 1997*), we tested the effect of gbz on the firing of BLA PNs in response to LEC stimulation (*Figure 1F–H*). However, $GABA_A$ receptor antagonism might increase the firing of BLA PNs by bringing them closer to AP threshold via membrane depolarization. To control for $GABA_A$ receptor blockade induced depolarization of the recorded PNs, we maintained the membrane voltage ($V_m$) of the PNs at −60 mV throughout these experiments (*Figure 1H*; p=0.38, Wilcoxon signed rank test). gbz mediated blockade of FFI led to a significant increase in PN firing in response to LEC stimulation (*Figure 1F,G*; p=0.031, Wilcoxon signed rank test). These findings confirm that the horizontal slice preparation contained a functionally complete LEC→BLA circuit supporting its use for the study of FFI.

## Cell type specificity of the LEC→BLA circuit

We performed minimal stimulation experiments in the LEC→BLA circuit using whole cell patch clamp electrophysiology to examine putative unitary synaptic events between neurons (*Gabernet et al., 2005*; *Kumar and Huguenard, 2001*). Briefly, stimulation intensity was tuned to a threshold intensity where LEC stimulation elicits eEPSCs in BLA neurons with a ~ 50% success rate and an all-or-none amplitude (*Figure 2—figure supplement 1*); at lower stimulation intensities EPSCs do not occur. As these EPSCs likely represent the response of the neuron to the activation of a single LEC neuron, this method provides a reliable measure of the unitary EPSC (uEPSC) between the stimulated and recorded neurons (*Gabernet et al., 2005*; *Kumar and Huguenard, 2001*). To identify dendritic-targeting Sst$^+$ and perisomatic-targeting PV$^+$ INs in acute brain slices, we generated Sst-tdTomato and PV-tdTomato mouse lines by crossing Sst-*ires*-Cre and PV-*ires*-Cre mice to *Ai9* tdTomato reporter line. Sst-tdTomato mice showed reliable labeling of Sst$^+$ cells and low overlap with PV$^+$ cells and PV-tdTomato mice showed reliable labeling of PV$^+$ cells and low overlap with Sst$^+$ cells (*Figure 2—figure supplement 2*). When minimally stimulating LEC we found that, whereas only a subset of BLA PNs (12/24 cells) and Sst$^+$ INs (16/35 cells) responded with EPSCs, nearly every PV$^+$ IN (15/16 cells) responded to the stimulation with an EPSC (*Figure 2A,B*). DNQX and D-APV abolished eEPSCs in all cell types, confirming the glutamatergic nature of the response (*Figure 2—figure supplement 1*).

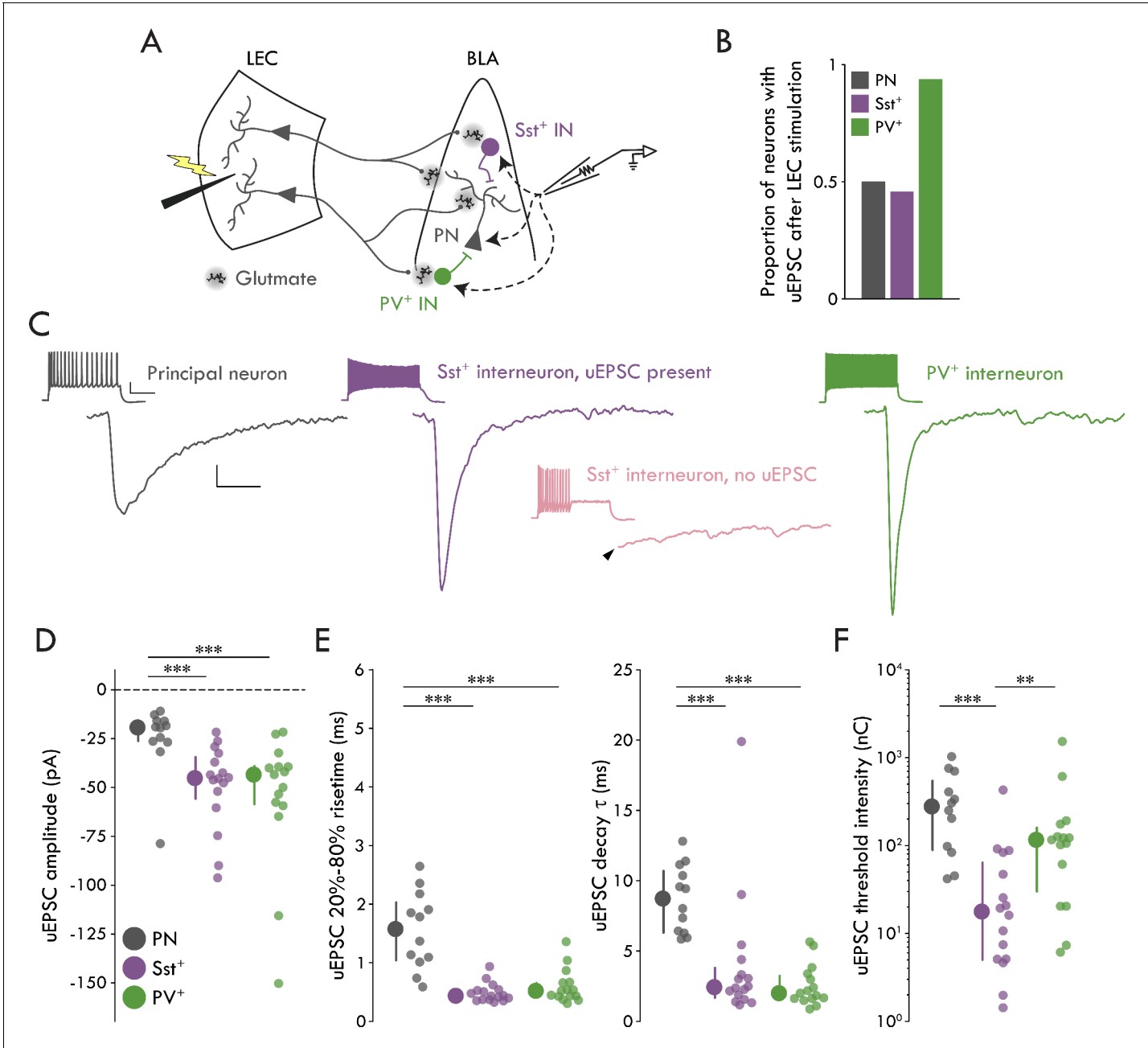

**Figure 2.** Cell type specificity of the LEC→BLA circuit. (**A**) Experimental schematic. (**B**) Proportion of neurons with detectable uEPSCs following LEC stimulation. (**C**) Mean traces of uEPSC successes from representative neurons. Left to right: PN (gray), Sst$^+$ IN with a detectable uEPSC (purple), PV$^+$ IN (green). Lower middle: mean trace of lack of response in a representative Sst$^+$ IN without a detectable uEPSC (pink). Arrowhead: truncated stimulus artifact. Scale bars: 5 pA, 5 ms. Insets: maximal firing to a current injection; scale bars: 20 mV, 200 ms. (**D**) uEPSC amplitude is larger in Sst$^+$/PV$^+$ INs compared to PNs (Kruskal-Wallis test: $p = 5.55 \times 10^{-4}$). (**E**) Left: uEPSC 20%-80% risetime is faster in Sst$^+$/PV$^+$ INs compared to PNs (Kruskal-Wallis test: $p = 1.66 \times 10^{-5}$). Right: uEPSC $\tau_{Decay}$ is faster in Sst$^+$/PV$^+$ INs compared to PNs (Kruskal-Wallis test: $p = 2.18 \times 10^{-5}$). (**F**) uEPSC threshold intensity is lower for Sst$^+$ INs (Kruskal-Wallis test: $p = 3.33 \times 10^{-4}$). Summary statistics in D, E, and F presented as median with IQR. **p<0.01, ***p<0.001. For all statistical tests: $n_{Sst}$ = 16, 8, $n_{PV}$ = 15, 5, $n_{PN}$ = 12, 8. See *Figure 2—figure supplement 1* for example minimal stimulation experiments and data showing the glutamatergic nature of the LEC→BLA synapse. See *Figure 2—figure supplement 2* for validation of the IN reporter mouse lines. See *Figure 2—figure supplement 3* for additional characterization of the LEC→BLA circuit. See *Figure 2—source data 1* for a table with full details on all statistical tests used in this figure. See *Figure 2—source data 2* for a table of all individual data points displayed in *Figure 2* and corresponding figure supplements.

The online version of this article includes the following source data and figure supplement(s) for figure 2:

**Source data 1.** Table of statistical analyses used in *Figure 2*.

*Figure 2 continued on next page*

*Figure 2 continued*

**Source data 2.** Table of data included in *Figure 2* and Supplements.
**Figure supplement 1.** Representative minimal stimulation experiments and glutamatergic nature of responses to LEC afferents.
**Figure supplement 2.** Validation of PV-tdTomato and Sst-tdTomato mouse lines.
**Figure supplement 3.** Further characterization of the LEC→BLA circuit.

Next, we compared putative uEPSCs across cell types (*Figure 2C-F*). We found that the uEPSC amplitude was larger in Sst⁺ and PV⁺ INs compared to PNs (*Figure 2D*; $p = 5.55 \times 10^{-4}$, Kruskal Wallis test). Additionally, we found the uEPSC kinetics were faster in the INs compared to PNs (*Figure 2E*; 20%-80% risetime: $p = 1.66 \times 10^{-5}$, Kruskal Wallis test; $\tau_{Decay}$: $p = 2.18 \times 10^{-5}$, Kruskal Wallis test). Finally, we found no differences between cell types in uEPSC latency or jitter (*Figure 2—figure supplement 3*). Together with our data demonstrating that BLA PNs rarely fire in response to LEC stimulation (*Figure 1E, F*), the statistically indistinguishable low jitters are consistent with a monosynaptic connection between LEC projection neurons and BLA Sst⁺ INs, PV⁺ INs, and PNs (*Doyle and Andresen, 2001*).

Since minimal stimulation likely reports the response of a BLA neuron to a single upstream LEC neuron (*Gabernet et al., 2005*; *Kumar and Huguenard, 2001*), we can use threshold stimulation intensity as an indirect measure of convergence of LEC projection neurons onto different BLA cell types. If threshold stimulation is lower for one BLA neuronal population compared to the others, it would follow that convergence of LEC inputs onto that cell type is likely greater relative to the other populations as it requires the activation of fewer LEC neurons. We found that threshold intensity was lower for the Sst⁺ INs compared to other cell types (*Figure 2F*; $p = 3.33 \times 10^{-4}$, Kruskal Wallis test).

Taken together, these data demonstrate that LEC stimulation leads to large and fast unitary currents in BLA Sst⁺ and PV⁺ INs and small and slow unitary currents in BLA PNs. Finally, although Sst⁺ and PV⁺ INs display equivalent uEPSCs, the finding that Sst⁺ INs have a lower threshold stimulation intensity compared to PV⁺ INs and PNs suggest that LEC afferents may have a greater functional convergence onto BLA Sst⁺ compared to PV⁺ INs.

## A fast spiking phenotype distinguishes BLA Sst⁺ INs targeted by LEC afferents

Though little is known about BLA Sst⁺ INs, they appear to have diverse electrophysiological properties ex vivo (*Krabbe et al., 2018*; *Sosulina et al., 2010*) and responses to stimuli in vivo (*Krabbe et al., 2018*; *Wolff et al., 2014*), suggesting Sst may be expressed by a broad range of GABAergic IN subtypes, similar to cerebral cortex (*Tremblay et al., 2016*). Only a subset of Sst⁺ INs responded to cortical stimulation raising the question whether these cells represented a distinct cell type. To address this hypothesis, we ran an unsupervised cluster analysis using Ward's method (*Ward, 1963*) based on 15 membrane properties from 105 Sst⁺ INs. Applying Thorndike's procedure (*Thorndike, 1953*) suggested two distinct clusters (Groups I and II). A large majority of Sst⁺ INs that responded to cortical stimulation clustered into Group I (14/16 cells) whereas a large majority of non-responsive Sst⁺ INs clustered into Group II (17/19 cells) (*Figure 3A*). To better understand what membrane properties best distinguished Group I and II Sst⁺ INs, we used decision tree analysis (*Breiman et al., 1984*; *Therneau and Atkinson, 1984*) which returned maximum firing rate, hyperpolarization induced sag, and AHP latency as the most salient parameters for cluster separation (*Figure 3—figure supplement 1*). To reduce potential model overfitting with the decision tree analysis, we used an additional predictive modeling technique, the random forest method (*Liaw and Wiener, 2002*). Like decision tree analysis, the random forest method returned maximum firing rate and hyperpolarization induced sag in addition to AP halfwidth as the most salient parameters for defining BLA Sst⁺ INs (*Figure 3B,C*).

Having observed two subpopulations of Sst⁺ INs, we compared membrane properties across Group I and II Sst⁺ INs and PV⁺ INs (*Figure 3D–F*, *Figure 3—source data 1*). These data show that Group I Sst⁺ and PV⁺ INs differed from Group II Sst⁺ INs in a subset of membrane properties such as maximum firing rate and hyperpolarization induced sag. Further, all three IN subtypes differed in AP halfwidth with PV⁺ INs firing the fastest APs and Group I Sst⁺ INs firing faster APs than Group II Sst⁺ INs (see *Figure 3—source data 1* for detailed statistics on membrane properties). The

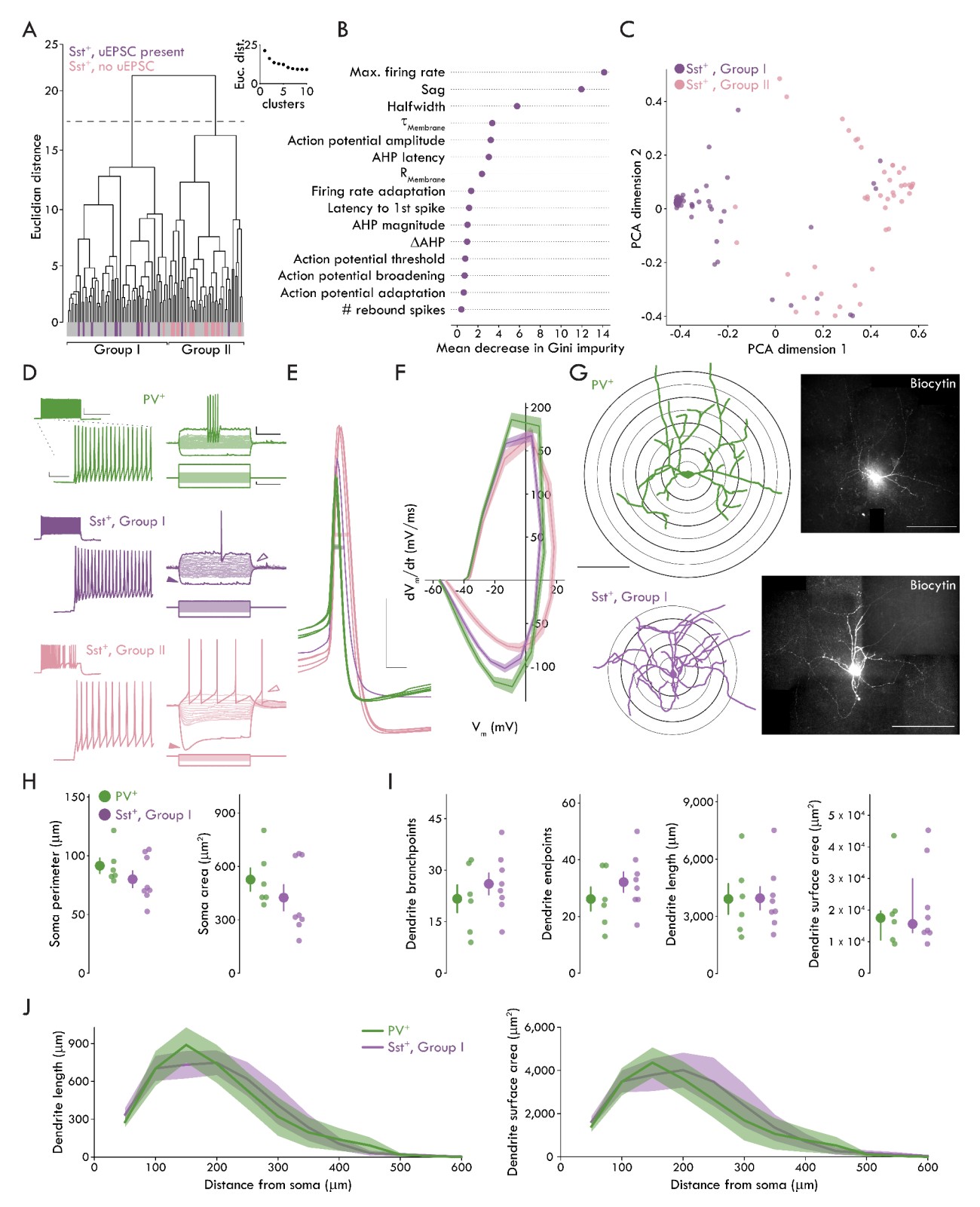

**Figure 3.** A fast spiking phenotype distinguishes BLA Sst[+] INs targeted by LEC afferents. (**A**) Unsupervised clustering analysis revealed two clusters of Sst[+] INs. Majority of Sst[+] INs with a uEPSC present following LEC stimulation cluster in Group I (14/16, Sst[+] INs with uEPSC; 60/105, all Sst[+] INs). Majority of Sst[+] INs with no response to LEC stimulation cluster in Group II (17/19, Sst[+] INs without uEPSC; 45/105, all Sst[+] INs). (**B**) Results of Random Forest model. Plot shows mean decrease in Gini impurity for each of the 15 parameters included in the model ($n_{Sst}$ = 105, 23). Gini impurity measures

*Figure 3 continued on next page*

*Figure 3 continued*

how often a random Sst⁺ IN would be clustered incorrectly if labeled randomly according to the distribution of labels in the set. (C) First two dimensions of the principal components analysis (PCA) of the random forest proximity matrix (n$_{Sst}$ = 105, 23). (D) Left: maximum AP firing from representative PV⁺ (top, green), Group I (middle, purple) and II (bottom, pink) Sst⁺ INs. Scale bars, main traces: 5 mV, 50 ms; inset: 20 mV, 400 ms. Right: voltage responses of the same neurons. Darker traces show responses to −200 pA and rheobase current injections. Lighter traces show responses to intermediate current injections were used to determine membrane resistance (−100 pA to sweep immediately before rheobase, Δ10 pA each sweep, +100 pA maximum current injection). Note lack of sag, rebound AP in Group I Sst⁺ IN (sag: closed arrowheads; rebound AP: open arrowheads). Scale bars: 10 mV, 200 ms; 50 pA, 200 ms. (E) APs of the same representative PV⁺ (green), Group I (purple), and II (pink) Sst⁺ INs at rheobase. Halfwidth: bar through width of AP. Scale bars: 20 mV, 1 ms. (F) Phase plots of PV⁺, Group I, and Group II Sst⁺ INs shows rate of voltage change for Group I and II APs at rheobase. Data presented as mean ± s.e.m. n$_{PV}$ = 52, 19, n$_{SstI}$ = 60, 23, n$_{SstII}$ = 45, 18. (G) Left: reconstructed soma and dendrites of a representative PV⁺ and Group I Sst⁺ INs (scale bar: 200 µm). Sholl rings shown beneath reconstruction. Right: confocal images of the biocytin filled PV⁺ and Group I Sst⁺ INs (scale bar: 200 µm). (H) No significant differences between PV⁺ and Group I Sst⁺ INs for soma perimeter (left; unpaired t-test: p=0.26) or area (right; Mann-Whitney U test: p=0.28). (I) No significant differences between PV⁺ and Group I Sst⁺ INs for (left to right) dendrite branchpoints (unpaired t-test: p=0.40), endpoints (unpaired t-test: p=0.30), length (unpaired t-test: p=0.97), or surface area (Mann-Whitney U test: p=0.95). (J) Results of Sholl analysis showing dendrite length (left) and surface area (right) as a function of distance from the soma in PV⁺ and Group I Sst⁺ INs. Data presented as mean ± s.e.m. n$_{PV}$ = 6, 4, n$_{SstI}$ = 8, 6. Summary statistics in H (perimeter) and I (branchpoints, endpoints, and length) presented as mean ± s.e.m. Summary statistics in H (area) and I (area) presented as median and IQR. For all statistical tests in (H) and (I): n$_{PV}$ = 6, 4, n$_{SstI}$ = 8, 6. See *Figure 3—figure supplement 1* for decision tree data. See *Figure 3—figure supplement 2* for data on Group II Sst⁺ IN morphology. See *Figure 3—source data 1* for a table of summary data and statistical comparisons on all membrane properties studied for Group I and Group II Sst⁺ INs and PV⁺ INs. See *Figure 3—source data 2* for a table with full details on all statistical tests used in this figure. See *Figure 3—source data 3* for a table of the results of the hierarchical cluster analysis and used by the Decision Tree and Random Forest models to determine which membrane properties best distinguished the clusters (related to *Figure 3A-C* and *Figure 3—figure supplement 1*). See *Figure 3—source data 4* for all individual data points of BLA IN membrane properties (related to *Figure 3—source data 1*). See *Figure 3—source data 5* for all individual data points for phase plot in *Figure 3F*. See *Figure 3—source data 6* for all individual data points on BLA IN morphology (related to *Figure 3H-J*). The online version of this article includes the following source data and figure supplement(s) for figure 3:

**Source data 1.** Differences in active and passive membrane properties among BLA Sst⁺ and PV⁺ INs.
**Source data 2.** Table of statistical analyses used in *Figure 3*.
**Source data 3.** Table of results of hierarchical clustering.
**Source data 4.** Table of membrane properties for all IN subtypes.
**Source data 5.** Table of membrane voltage and derivative of membrane voltage, related to *Figure 3F*.
**Source data 6.** Table of morphological data for *Figure 3* and Supplements.
**Figure supplement 1.** Decision tree analysis to determine most salient parameters that discriminate Group I and II Sst⁺ INs.
**Figure supplement 2.** Group II Sst⁺ IN morphology properties.

membrane properties of Group I Sst⁺ and PV⁺ INs are consistent with a FS phenotype typically seen in cortical, hippocampal, and BLA PV⁺ INs that is characterized by the ability to fire high frequency trains of brief APs (*Tremblay et al., 2016*; *Woodruff and Sah, 2007a*; however, see *Large et al. (2016)*; *Ma et al. (2006)*; *Nigro et al. (2018)* for examples of FS Sst⁺ INs in cerebral cortex).

In order to compare the morphology of the IN populations, we included biocytin in the patch pipette in a subset of recordings to allow for *post hoc* visualization of the different IN subtypes. Using these biocytin-filled neurons, we created reconstructions of the somatic and dendritic morphology of the INs (*Figure 3G*; see *Figure 3—figure supplement 2* for Group II Sst⁺ IN data which were not included in formal analyses due to low number of recovered morphologies [n$_{SstII}$ = 3, 2]). When we compared the somatic and dendritic morphology of PV⁺ and Group I Sst⁺ INs, we found no significant differences in any measurement (*Figure 3H–J*, see *Figure 3—source data 2* for statistical analysis).

Taken together, the membrane properties of BLA Sst⁺ and PV⁺ INs reveal two distinct subpopulations of Sst⁺ INs that are readily distinguished at the biophysical level by their FS phenotype and at the functional circuit level by synaptic responses to cortical stimulation. However, the IN subtypes do not appear to have any readily observable differences in their somatic and dendritic morphology. For clarity, we refer to the Group I and II Sst⁺ INs as FS and non-fast spiking (nFS) Sst⁺ INs, respectively.

## Probing the LEC→BLA circuitry suggests distinct functional feedforward/feedback roles for IN subtypes

BLA Sst$^+$ INs have lower threshold stimulation intensity compared to PV$^+$ INs and PNs (*Figure 2F*). Since these data suggest a higher rate of convergence of LEC afferents onto Sst$^+$ compared to PV$^+$ INs, we wanted to test the hypothesis that LEC input to BLA may preferentially recruit Sst$^+$ over PV$^+$ INs. To do this, we recorded from BLA Sst$^+$ and PV$^+$ INs in current clamp at rest and stimulated LEC with five pulses at 20 Hz. The stimulation intensity was set to the empirically derived threshold stimulation for BLA PNs (defined as the median PN threshold stimulation; 273.00 nC; *Figure 2F*) allowing us to determine how BLA INs respond when LEC activity is sufficient to ensure that PNs receive input. We found that stimulation led to robust spiking in BLA Sst$^+$ INs whereas PV$^+$ INs rarely fired (*Figure 4A–C*; p=0.016, Mann-Whitney U test).

To probe the underlying mechanisms of the preferential spiking of Sst$^+$ INs, we used two additional measures: resting $V_m$ ($V_{rest}$) and subthreshold EPSP amplitude. We found that $V_{rest}$ was more depolarized in Sst$^+$ compared to PV$^+$ INs (*Figure 4D*; $p = 0.034$, unpaired t-test). Because individual INs displayed variability in the number of APs/stimulus, each IN would have a different number of subthreshold EPSPs. Therefore, we compared both the distribution of all subthreshold EPSP amplitudes and the mean cellular amplitudes across Sst$^+$ and PV$^+$ INs. Regardless of the analysis method, we found that evoked subthreshold EPSPs were larger in Sst$^+$ compared to PV$^+$ INs (*Figure 4E*; all subthreshold EPSP events: $p = 2.19 \times 10^{-21}$, Kolmogorov-Smirnov test; cellular subthreshold EPSP: $p = 0.030$, Mann-Whitney U test). Thus, the data show that two distinct mechanisms underlie the recruitment of BLA Sst$^+$ INs by LEC afferents: Sst$^+$ INs were more depolarized at rest compared to PV$^+$ INs, and, despite the concomitant decrease in the driving force through glutamate receptors, the magnitude of evoked subthreshold EPSPs was greater in Sst$^+$ compared to PV$^+$ INs. Finally, when we classified the Sst$^+$ INs from this experiment (*Figure 3A*), we found that all but one (7/8 cells) were FS Sst$^+$ INs, consistent with our finding that LEC afferents appear to selectively target FS cells among the Sst$^+$ INs.

Taken together with the data in *Figures 2* and *3*, these data show that LEC stimulation leads to synaptic responses in BLA FS Sst$^+$ INs, PV$^+$ INs, and PNs but not in nFS Sst$^+$ INs. Although FS Sst$^+$ and PV INs have equivalent unitary events following LEC stimulation, the responses of these cell types to LEC input diverge in important ways. Specifically, FS Sst$^+$ INs have a lower threshold stimulation intensity compared to PV$^+$ INs, and FS Sst$^+$ INs have greater subthreshold EPSP amplitudes and are more likely to fire in response to PN threshold stimulation of LEC compared to PV$^+$ INs. These findings suggest that LEC afferents to BLA have a greater functional convergence onto FS Sst$^+$ INs compared to PV$^+$ INs and raise the question whether they are involved in BLA FFI.

In neocortical circuits, PV$^+$ INs mediate FFI whereas Sst$^+$ typically mediate feedback inhibition (*Tremblay et al., 2016*); nevertheless, our data point to a role for Sst$^+$ INs in BLA FFI, consistent with prior hypotheses that BLA Sst$^+$ INs mediate FFI and PV$^+$ INs mediate feedback inhibition (*Duvarci and Pare, 2014*). Thus, we wanted to test for responses of Sst$^+$ and PV$^+$ INs to local BLA PNs to probe for potential roles in feedback inhibition. To do this, we recorded PN-IN pairs in BLA, drove AP firing in PNs and compared uEPSC responses to the AP across IN subtypes (*Figure 4F–H*). We found responses in all IN subtypes (8/10 FS Sst$^+$, 5/5 nFS Sst$^+$, 8/12 PV$^+$ INs with uEPSCs in response to PN APs) and found that BLA PV$^+$ INs responded to PNs with larger amplitude uEPSCs compared to either FS or nFS Sst$^+$ INs (*Figure 4H*; p=0.025, Kruskal-Wallis test). Further, we found no difference in the latency or jitter of the uEPSCs across cell types (*Figure 4—figure supplement 1*). These data are consistent with a larger role for PV$^+$ INs in BLA feedback inhibition relative to Sst$^+$ INs.

Finally, we wanted to look at the level of spontaneous excitation onto the different IN subtypes as different levels of spontaneous glutamatergic activity could lead to different levels of basal excitability for the INs in the LEC→BLA circuit (*Figure 4I,J*). We recorded spontaneous EPSCs (sEPSCs) across BLA IN subtypes. We found that PV$^+$ INs had larger amplitude sEPSCs than either Sst$^+$ IN subtype and that PV$^+$ and FS Sst$^+$ INs had more frequent sEPSCs than nFS Sst$^+$ INs (*Figure 4K*; amplitude: p=0.022, one-way ANOVA; frequency: p=0.022, one-way ANOVA). Additionally, sEPSCs in nFS Sst$^+$ INs had slower decay kinetics compared to the other BLA IN subtypes (*Figure 4L*, *Figure 4—figure supplement 1*; $\tau_{Decay}$: p=0.0033, Kruskal-Wallis test). These data suggest that although FS Sst$^+$ and PV$^+$ INs may have divergent functional roles with regards to feedforward and

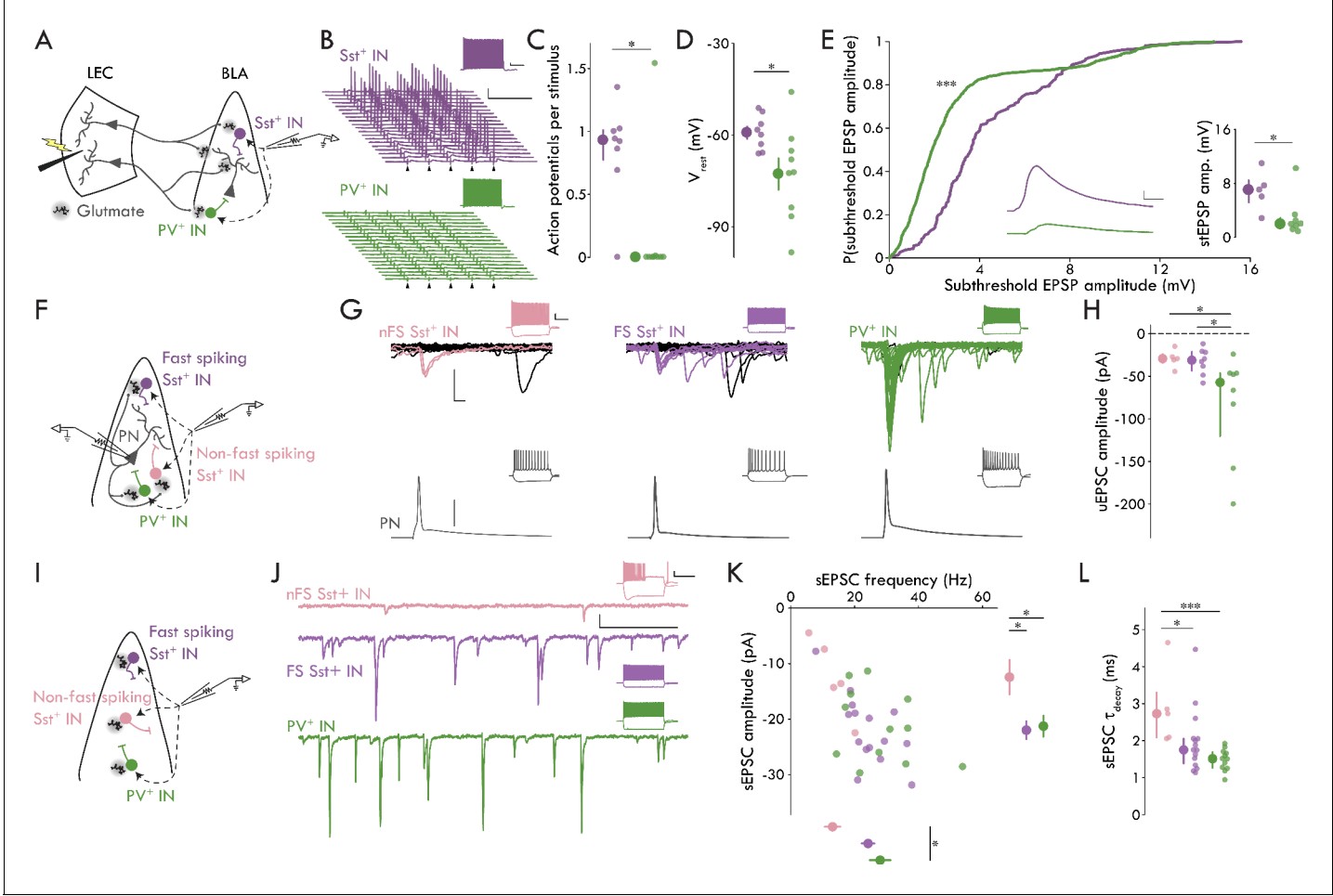

**Figure 4.** Distinct microcircuit functional roles for IN subtypes. (A) Experimental schematic for B-E. (B) Representative Sst$^+$ (top, purple) and PV$^+$ (bottom, green) responses to LEC stimulation at 20 Hz (arrowhead: stimulation artifacts; scale bars: 25 mV, 100 ms). Insets show maximal firing frequency for the representative IN in response to a square current pulse (scale bars: 5 mV, 200 ms). (C) Sst$^+$ INs fire more APs per stimulus compared to PV$^+$ INs (Mann-Whitney U test: p=0.016; $n_{Sst}$ = 8, 3; $n_{PV}$ = 9, 3). (D) Sst$^+$ INs have a more depolarized $V_{rest}$ compared to PV$^+$ INs (unpaired t-test: p=0.034; $n_{Sst}$ = 8, 3; $n_{PV}$ = 9, 3). (E) Cumulative probability distribution of all subthreshold EPSPs. Inset, middle: mean subthreshold EPSP from representative neurons in (A; Sst$^+$: purple, PV$^+$: green; scale bars: 1 mV, 5 ms; Kolmogorov-Smirnov test: $p$ = 2.19 x 10$^{-21}$, $n_{Sst}$ = 161 events, $n_{PV}$ = 779 events; Mann-Whitney U test: $p$ = 0.030, $n_{Sst}$ = 5, 3, $n_{PV}$ = 8, 3). (F) Experimental schematic for G and H. (G) Representative paired recording experiments between BLA and INs (left to right: nFS Sst$^+$, FS Sst$^+$, PV$^+$). Bottom: AP in BLA PN (scale bar: 40 mV). Top: overlaid current responses of BLA INs to PN AP (50 pA, 2 ms); successful trials shown in color. Insets show maximal firing frequency and response to a −200 pA current injection in the representative neurons (scale bars: 20 mV, 200 ms). (H) uEPSC amplitude is larger in PV$^+$ compared to FS and nFS Sst$^+$ INs (Kruskal-Wallis test test: p=0.025, $n_{nFS-Sst}$ = 5, 5, $n_{FS-Sst}$ = 8, 6, $n_{PV}$ = 8, 8) (I) Experimental schematic for J-L. (J) Representative sEPSC traces from BLA INs (top to bottom: nFS Sst$^+$, FS Sst$^+$, PV$^+$; scale bars: 20 pA, 100 ms). Insets show maximal firing frequency and response to a −200 pA current injection in the representative neurons (scale bars: 20 mV, 300 ms). (K) Scatter plot of sEPSC frequency and amplitude. Right: nFS Sst$^+$ INs have smaller amplitude sEPSCs compared to FS Sst$^+$ and PV$^+$ INs (one-way ANOVA: p=0.022). Bottom: PV$^+$ INs have more frequent sEPSCs compared to nFS Sst$^+$ INs (one-way ANOVA: p=0.022). (L) nFS Sst$^+$ IN sEPSCs have slower decay kinetics compared to FS Sst$^+$ and PV$^+$ IN sEPSCs (Kruskal-Wallis test: p=0.0033). Summary statistics in D and K presented in color as mean ± s.e.m. and in C, E (inset), H, and L in color as median with IQR. *p < 0.05 or False Discovery Rate corrected significance threshold where applicable, ***p < 0.001. See *Figure 4—figure supplement 1* for additional characterization of the BLA inhibitory microcircuitry. See *Figure 4—source data 1* for a table with full details on all statistical tests used in this figure. See *Figure 4—source data 2* for a table of all individual data points displayed in *Figure 4* and corresponding figure supplements.

The online version of this article includes the following source data and figure supplement(s) for figure 4:

**Source data 1.** Table of statistical analyses used in *Figure 4*.
**Source data 2.** Table of data included in *Figure 4* and Supplements.
**Figure supplement 1.** Further characterization of BLA inhibitory microcircuitry.

feedback inhibition, they both receive greater levels of spontaneous excitatory input compared to nFS Sst$^+$ INs.

## Sst$^+$ INs mediate LEC→BLA FFI

To test whether Sst$^+$ or PV$^+$ INs provide cortically evoked FFI onto BLA PNs, we generated Sst-hM4Di and PV-hM4Di mice by crossing the Sst-*ires*-Cre or PV-*ires*-Cre mice to the ROSA-hM4Di-mCitrine mice. These mice selectively express hM4Di in Sst$^+$ (Sst-hM4Di) or PV$^+$ (PV-hM4Di) INs. hM4Di is an inhibitory chemogenetic receptor that, when bound to its ligand clozapine-*N*-oxide (CNO), activates the G$_{i/o}$ signaling cascade to hyperpolarize and block neurotransmitter release from neurons (*Armbruster et al., 2007*; *Stachniak et al., 2014*). To validate the efficacy of the Sst-hM4Di and PV-hM4Di mouse lines, we patched onto BLA mCitrine$^+$ neurons and recorded V$_m$ in current

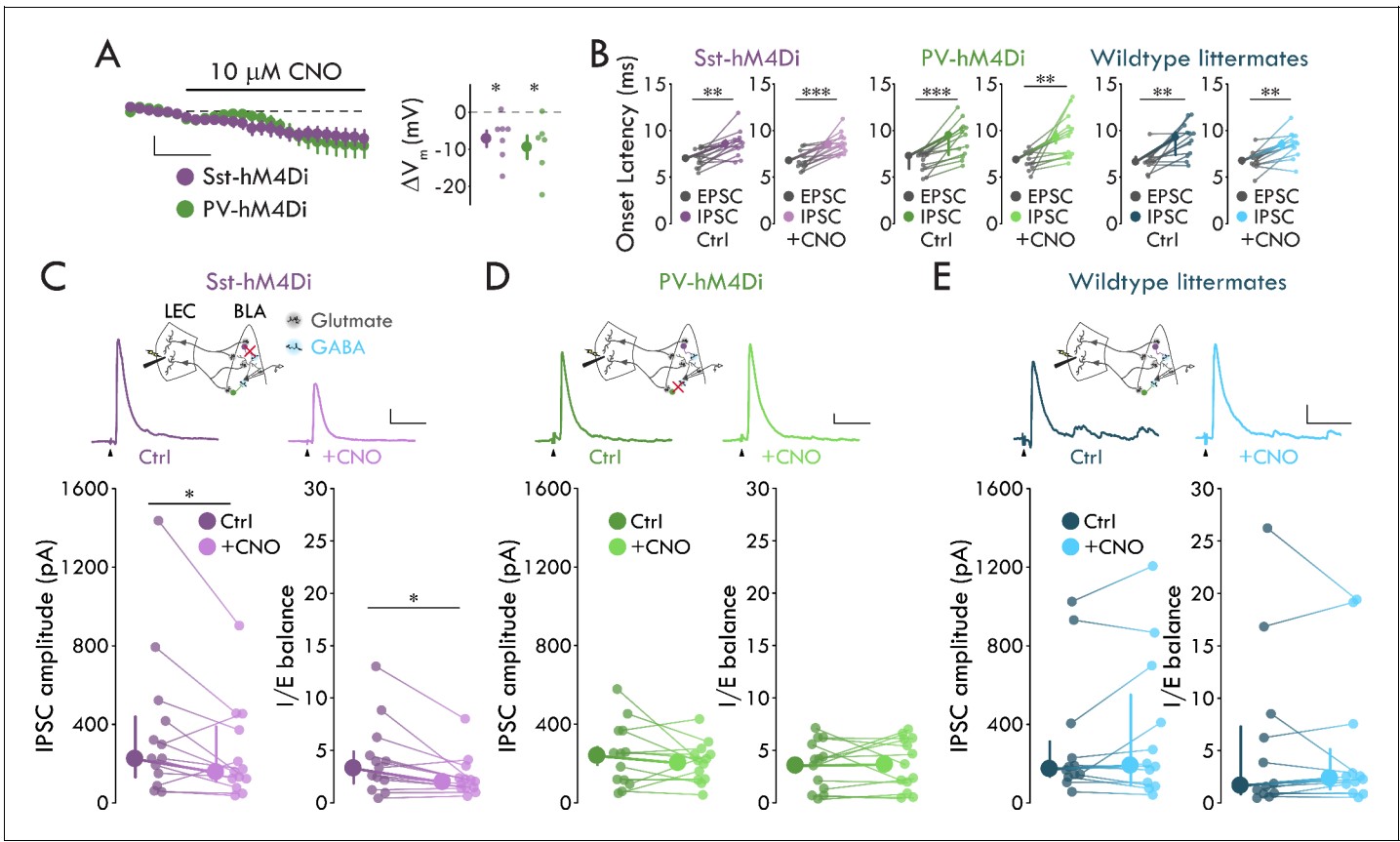

**Figure 5.** Sst$^+$ INs mediate LEC→BLA FFI. (**A**) Left: Effect of CNO on V$_m$. Scale bars: 5 mV, 3 min. Right: CNO hyperpolarizes mCitrine$^+$ cells in both mouse lines ($p_{Sst}$ = 0.017, $p_{PV}$ = 0.031, one-sample t-test, $n_{Sst}$ = 7, 4, $n_{PV}$ = 6, 3). (**B**) IPSC is delayed relative to EPSC (paired t-tests: $p_{Sst-Ctrl}$ = 0.0018, $p_{Sst-CNO}$ = 8.80 x 10$^{-4}$, $p_{PV-CNO}$ = 0.0042, $p_{WT-CNO}$ = 0.0012; Wilcoxon signed-rank tests: $p_{PV-Ctrl}$ = 2.44 x 10$^{-4}$, $p_{WT-Ctrl}$ = 0.0024; $n_{Sst}$ = 13, 6, $n_{PV}$ = 13, 7, $n_{WT}$ = 12, 6). (**C**) CNO- mediated Sst$^+$ IN inactivation reduces FFI (Sst-hM4Di mouse line; Wilcoxon signed-rank tests: $p_{IPSC}$ = 0.040, $p_{IE}$ = 0.013, n = 13, 6). Top: Experimental schema, IPSC traces with and without CNO. Arrowheads: stimulation artifacts (truncated). Scale bars: 100 pA, 50 ms. (**D**) PV$^+$ IN inactivation has no effect on FFI (paired t-tests: $p_{IPSC}$ = 0.35, $p_{IE}$ = 0.85, n = 13, 7). Display as for (**C**), but in PV-hM4Di mouse line. Scale bars: 25 pA, 50 ms. (**E**) CNO has no effect on FFI in wildtype littermates (Wilcoxon signed-rank tests: $p_{IPSC}$ = 0.73, $p_{IE}$ = 0.68, n = 12, 6). Display as for (**C**), but in hM4Di$^{-/-}$ littermates of Sst- and PV-hM4Di mice. Scale bars: 50 pA, 50 ms. Summary statistics in A, B (Sst-hM4Di both conditions, PV-hM4Di CNO condition, WT littermates CNO condition), and D in color as mean ± s.e.m. Summary statistics in B (PV-hM4Di and WT littermates control conditions), C, and E in color as median with IQR. *$p < 0.05$, **$p < 0.01$, ***$p < 0.001$. See *Figure 5—figure supplement 1* for lack of effect of IN inactivation on EPSCs. See *Figure 5—source data 1* for a table with full details on all statistical tests used in this figure. See *Figure 5—source data 2* for a table of all individual data points displayed in *Figure 5* and corresponding figure supplements.

The online version of this article includes the following source data and figure supplement(s) for figure 5:

**Source data 1.** Table of statistical analyses used in *Figure 5*.
**Source data 2.** Table of data included in *Figure 5* and Supplements.
**Figure supplement 1.** Lack of an effect of 10 μM CNO on evoked EPSC amplitude.

clamp ($I_{hold}$ = 0 pA). Application of 10 µM CNO reduced $V_m$ of mCitrine$^+$ neurons in both lines (*Figure 5A*; $p_{Sst}$ = 0.017, $p_{PV}$ = 0.031, one-sample t-test). Thus, we used the Sst-hM4Di and PV-hM4Di mice to selectively hyperpolarize Sst$^+$ or PV$^+$ INs.

To assess the role of Sst$^+$ and PV INs in BLA FFI, we stimulated LEC and recorded evoked EPSCs and IPSCs in BLA PNs using Sst-hM4Di and PV-hM4Di mice before and after bath application of CNO (10 µM). To determine the effect of IN inactivation on FFI we quantified two measures: evoked IPSC amplitude and inhibition-excitation balance (I/E balance). We defined I/E balance as the evoked IPSC amplitude for each cell normalized to its evoked EPSC amplitude. We examined this measure in addition to evoked IPSC amplitude to control for potential differences across cells with regards to the number of excitatory inputs to BLA activated by LEC stimulation. Consistent with the disynaptic nature of LEC-driven FFI in BLA (*Figure 1*), LEC stimulation elicited an IPSC that was significantly delayed relative to the EPSC in BLA PNs in both Sst-hM4Di and PV-hM4Di mice in control conditions and in the presence of CNO (*Figure 5B*, see *Figure 5—source data 1* for detailed statistical analysis). Importantly, these data indicate that hM4Di expression or CNO application does not alter the ability of LEC stimulation to drive FFI. When we perfused CNO to inactivate the different IN subtypes, CNO reduced IPSC amplitude and I/E balance in PNs from Sst-hM4Di mice by 30.2% and 40.2% respectively but had no effect on either measure in PNs from PV-hM4Di mice (*Figure 5C,D*; $p_{Sst-IPSC}$ = 0.040, $p_{Sst-IE}$ = 0.013, Wilcoxon signed rank tests; $p_{PV-IPSC}$ = 0.35, $p_{PV-IE}$ = 0.85, paired t-tests). Finally, demonstrating the effects of Sst$^+$ and PV$^+$ IN inactivation were specific to FFI, bath application of CNO had no effect on eEPSCs in either mouse line (*Figure 5—figure supplement 1*).

To control for off-target effects of CNO (*Gomez et al., 2017*), we repeated the experiments in wildtype (WT) littermates of Sst-hM4Di and PV-hM4Di mice. In WT littermates, LEC stimulation elicited the delayed EPSC-IPSC pairing consistent with FFI (*Figure 5B*, see *Figure 5—source data 1* for detailed statistical analysis), and CNO application had no effect on LEC-driven excitation or FFI in BLA (*Figure 5E*, *Figure 5—figure supplement 1*, see *Figure 5—source data 1* for detailed statistical analysis). Taken together with our data demonstrating that LEC afferents selectively synapse onto FS Sst$^+$ INs among the Sst$^+$ IN subtypes (*Figures 2–3*), these data show that FS Sst$^+$, but not PV$^+$ or nFS Sst$^+$, INs mediate LEC-driven FFI in BLA.

## Discussion

Our data show that a previously unreported subpopulation of fast spiking Sst$^+$ INs mediates LEC-driven FFI in BLA (*Figure 6*). Although LEC afferents synapsed onto both PV$^+$ and FS Sst$^+$ INs (*Figures 2* and *3*), only FS Sst$^+$ INs fired in response to LEC activity (*Figure 4*); conversely, PV$^+$ INs received stronger synaptic input from local BLA PNs than both populations of Sst$^+$ INs. Finally, inactivation of Sst$^+$, but not PV$^+$, INs led to a reduction in FFI onto BLA PNs (*Figure 5*). Our findings are consistent with prior experiments in BLA indicating that Sst$^+$ INs form the anatomical circuit underlying FFI (*Duvarci and Pare, 2014*; *Smith et al., 2000*; *Unal et al., 2014*). In summary, these data illuminate a circuit mechanism for the feedforward inhibitory control mediated by entorhinal afferents onto BLA PNs (*Lang and Paré, 1997*). The role of BLA FFI in gating plasticity at afferent synapses onto PNs (*Bazelot et al., 2015*; *Bissière et al., 2003*; *Tully et al., 2007*) and of local Sst$^+$ INs in gating BLA-dependent learning in vivo (*Wolff et al., 2014*) suggests a potential role for this circuit in olfactory and other forms of sensory-valence learning (*Keene et al., 2016*; *Kitamura et al., 2017*; *McDonald, 1998*; *Mouly and Di Scala, 2006*; *Schoenbaum et al., 1999*; *Tsao et al., 2013*; *Xu and Wilson, 2012*).

BLA Sst$^+$ INs are a heterogenous population of dendritic targeting INs with diverse firing properties (*Krabbe et al., 2018*; *Muller et al., 2007*; *Sosulina et al., 2010*; *Wolff et al., 2014*). This is similar to what is observed in cerebral cortex (*Tremblay et al., 2016*). Accordingly, our cluster analysis and predictive modeling techniques revealed two distinct subpopulations of Sst$^+$ INs that could be distinguished based on membrane properties, most notably maximum firing rate, hyperpolarization induced sag, and AP halfwidth (*Figure 3*). These membrane properties suggest differential expression of HCN and $K_v3$ channels in the Sst$^+$ IN subpopulations (*Poolos et al., 2002*; *Rudy and McBain, 2001*). Consistent with this, $K_v3.2$ is expressed in a large subpopulation of BLA Sst$^+$ INs (*McDonald and Mascagni, 2006*).

How IN heterogeneity maps onto specific functions within the BLA is critical for understanding survival circuits. We observed that our cluster analysis based on intrinsic properties separated the

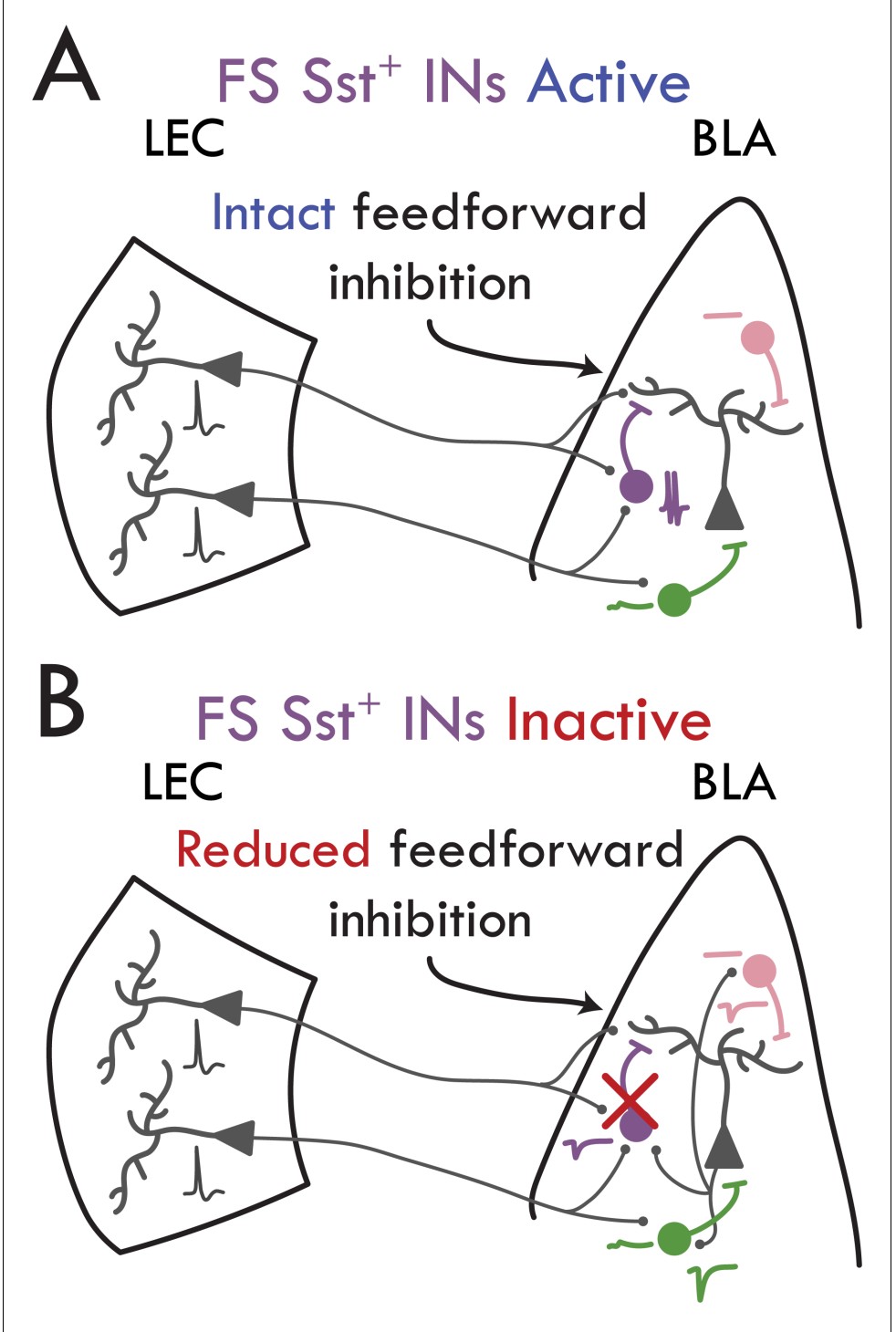

**Figure 6.** Schematic of findings. (**A**) FS Sst[+] INs (purple) fire APs in response to LEC input to drive FFI onto BLA PNs. nFS Sst[+] INs (pink) do not respond to LEC input. PV[+] INs (green) respond with a subthreshold excitatory response to LEC input. (**B**) Inactivation of FS Sst[+] IN removes the feedforward inhibitory control over BLA PNs. PV[+] INs receive large uEPSC input from local BLA PNs compared to both Sst[+] IN subtypes.

populations of Sst$^+$ INs based on whether they displayed a synaptic response to LEC stimulation: 87.5% of Sst$^+$ INs with a synaptic response were classified as fast spiking and 89.5% of Sst$^+$ without a synaptic response were non-fast spiking. Although canonically PV$^+$ INs are thought to be synonymous with FS INs, our data indicate that, at least in BLA, this is not the case. In BLA, our data reveal two functionally distinct populations of FS INs can be separated based on PV or Sst expression. Further, taken together with recent descriptions of FS Sst$^+$ INs in cortex (*Large et al., 2016*; *Ma et al., 2006*; *Naka et al., 2019*; *Nigro et al., 2018*), we argue that a fast spiking phenotype is insufficient to classify an IN solely as PV$^+$. Indeed, despite similar physiological phenotypes, in neocortex, FS Sst$^+$ INs have distinct functions from PV$^+$ INs. For example, whereas a well characterized circuit role of PV$^+$ INs is their control of thalamocortical driven FFI onto layer 4 PNs (*Tremblay et al., 2016*), a recent elegant study demonstrated that neocortical FS Sst$^+$ receive their input from layer 4 PNs and provide strong feedback inhibition to these PNs that is capable of silencing sensory evoked responses in the PNs (*Naka et al., 2019*). Further, in piriform and neocortex, FS Sst$^+$ INs show clear biophysical and circuit connectivity differences from other Sst$^+$ IN subtypes (*Large et al., 2016*; *Ma et al., 2006*; *Naka et al., 2019*; *Nigro et al., 2018*). Recordings and manipulations of neocortical Sst$^+$ IN subtypes in awake, behaving animals show that FS Sst$^+$ and nFS Sst$^+$ INs have disparate functioning in the neocortical circuit during sensory processing (*Muñoz et al., 2017*; *Naka et al., 2019*). Thus, our data are consistent with recent findings from cerebral cortex and demonstrate that FS Sst$^+$ INs are biophysically and functionally distinct class of Sst$^+$ INs. Further, we show that BLA FS Sst$^+$ INs mediate feedforward inhibition in the LEC→BLA circuit.

In cerebral cortex and hippocampus, a major circuit role of perisomatic PV$^+$ INs is to regulate AP output of PNs via FFI (*Glickfeld and Scanziani, 2006*; *Pouille and Scanziani, 2001*; *Tremblay et al., 2016*). Similarly in BLA, PV$^+$ INs provide strong regulation of local PN activity (*Andrási et al., 2017*; *Veres et al., 2017*; *Woodruff and Sah, 2007b*). Nevertheless, anatomical data indicates that BLA PV$^+$ INs receive more synaptic input from local PNs as compared to afferent sources (*Smith et al., 2000*). Our data demonstrating that PV$^+$ INs receive larger uEPSCs from local PNs compared to Sst$^+$ INs (*Figure 4*) and other recent work showing that PV$^+$ INs fire APs more readily in response to local PN activation compared to cholecystokinin expressing INs (*Andrási et al., 2017*) provide a physiological compliment to the anatomy findings. Thus, PV$^+$ INs are positioned to provide feedback and lateral inhibition to control BLA spiking output whereas FS Sst$^+$ INs provide feedforward inhibition to regulate PN responses to sensory afferents to BLA. Together, these inhibitory networks would work in concert to fine-tune patterns of BLA activity to allow animals to learn about and react to stimuli. Our results illuminate an important point: archetypal circuit motifs that reappear across brain regions may be realized by different types of neurons in these brain areas depending on the functional role of the local circuit motif.

Although our data indicate that BLA PV$^+$ INs do not fire in response to LEC stimulation, afferent activation does lead to postsynaptic responses in these INs (*Figures 2* and *4*). When considered with our finding that local PNs preferentially target this IN subtype (*Figure 4*), this finding suggests an intriguing possibility for PV$^+$ IN functioning. Specifically, if afferents drive specific subsets of BLA PNs to fire, this afferent driven subthreshold depolarization of PV$^+$ INs would prime them to be more likely to fire in response to the subsequent PN activity. As BLA PV$^+$ IN firing is sufficient to prevent AP firing in postsynaptic PNs (*Andrási et al., 2017*; *Veres et al., 2017*; *Woodruff and Sah, 2007b*), this motif could provide an important source of lateral inhibition to select for activity in specific subsets of BLA PNs in response to sensory stimuli. Indeed, in vivo electrophysiological and Ca$^{2+}$ imaging studies in behaving animals indicate that sensory stimuli drive activity in specific subpopulations of BLA PNs rather than widespread activity across BLA PNs (*Beyeler et al., 2016*; *Grewe et al., 2017*; *Schoenbaum et al., 1999*).

Do the findings of the LEC→BLA circuit extend to other afferents to BLA (e.g. thalamic and other cortical inputs)? A recent study (*Lucas et al., 2016*) shows that PV$^+$ INs mediate polysynaptic inhibition in the lateral but not basal subnucleus of the BLA following stimulation of cortical (external capsule) or thalamic (internal capsule) fiber tracts to BLA. As stimulation of these fiber tracts leads to activation of local PNs and feedback inhibition (*Szinyei et al., 2000*), their results could be indicative of PV$^+$ INs mediating feedback inhibition in the lateral subnucleus of the BLA, as has been suggested previously (*Duvarci and Pare, 2014*). The horizontal slice preparation does not allow for unambiguous discrimination between the lateral and basal subnuclei of the BLA; therefore, we cannot determine the subnucleus within BLA where we conducted our recordings. Since the majority of LEC

afferents target the basal subnucleus of the BLA (*McDonald, 1998*), it is conceivable that we recorded predominantly from neurons in this BLA subnucleus. If this were the case, this would provide an alternative explanation for the apparent discrepancy between our data and the data reported by Lucas and colleagues (*Lucas et al., 2016*); namely, that PV$^+$ INs provide FFI in the lateral subnucleus and Sst$^+$ INs provide FFI in the basal subnucleus. However, BLA PV$^+$ INs of both the lateral and basal subnuclei of the BLA receive relatively little input from a variety of cortical sources (*Smith et al., 2000*). Instead, cortical inputs target PV$^-$/calbindin$^+$ INs throughout the BLA (*Unal et al., 2014*). Interestingly, the major population of PV$^-$/calbindin$^+$ INs in BLA are Sst$^+$ INs (*McDonald and Mascagni, 2002*). Taken together these data suggest that our findings that FS Sst$^+$ INs mediate BLA FFI will likely hold across cortical inputs to the BLA. Nevertheless, future experiments will be necessary to test this premise.

What might be the functional role of BLA FS Sst$^+$ IN mediated FFI? In BLA, FFI plays a critical role in exerting control over the *plasticity* of PNs (*Bazelot et al., 2015*; *Bissière et al., 2003*; *Tully et al., 2007*). BLA Sst$^+$ INs primarily target the dendrites of PNs (*Muller et al., 2007*; *Wolff et al., 2014*). One function of dendritic targeting INs is to regulate the local Ca$^{2+}$ flux in dendrites (*Chiu et al., 2013*; *Miles et al., 1996*; *Müllner et al., 2015*). Recent work in cerebral cortex and BLA demonstrates that dendritic disinhibition via Sst$^+$ INs inhibition is critical in gating cortical pyramidal neuron plasticity ex vivo and in gating learning in vivo via regulation of learning dependent changes in pyramidal neuron activity (*Adler et al., 2019*; *Williams and Holtmaat, 2019*; *Wolff et al., 2014*). Specifically, sensory stimuli inhibit Sst$^+$ IN firing in BLA (*Wolff et al., 2014*), and further decreasing Sst$^+$ IN activity facilitates learning whereas increasing Sst$^+$ IN activity impairs learning (*Wolff et al., 2014*). Taken in the context of these findings, our data suggest an intriguing potential circuit mechanism explaining how this pattern of activity in vivo may lead to learning. Sensory stimulus mediated inhibition of Sst$^+$ INs would lead to a reduction in FFI and consequent dendritic disinhibition which could allow for plastic changes in the BLA PN to determine appropriate behavioral responses to the sensory stimulus in the future. Future experiments should test the hypothesis that Sst$^+$ IN mediated FFI of BLA PNs gates BLA plasticity and learning.

Taken together, the data presented in this study demonstrate the existence of a previously unreported population of FS Sst$^+$ INs in BLA. This population of INs is biophysically and functionally distinct from PV$^+$ and other Sst$^+$ INs. We show that these FS Sst$^+$ INs provide FFI at the corticoamygdalar synapse in contrast to nFS Sst$^+$ and PV$^+$ INs that do not engage in this circuit motif. Our data highlight the importance of probing IN heterogeneity within the context of local circuits as a key aspect for understanding circuit functioning. As FFI gates synaptic plasticity in BLA (*Bazelot et al., 2015*; *Bissière et al., 2003*; *Tully et al., 2007*), our data raise the possibility that the FFI mediated by this novel BLA population of FS Sst$^+$ INs serves as the underlying circuit mechanism for learning in the survival circuits of the amygdala.

# Materials and methods

### Key resources table

| Reagent type | Reagent or resource | Source | Identifier | Additional information |
|---|---|---|---|---|
| Antibody | Rat monoclonal anti-Sst | Millipore | Cat#MAB354; RRID: AB_2255365 | (1:100) |
| Antibody | Guinea pig polyclonal anti-PV | Synaptic Systems | Cat#195004; RRID: AB_2156476 | (1:500) |
| Antibody | Donkey polyclonal anti-rat Alexa Fluor 488 | Jackson ImmunoResearch | Cat#712-545-150 | (1:500) |
| Antibody | Donkey polyclonal anti-guinea pig Alexa Fluor 488 | Jackson ImmunoResearch | Cat#706-545-148 | (1:500) |

*Continued on next page*

*Continued*

| Reagent type | Reagent or resource | Source | Identifier | Additional information |
|---|---|---|---|---|
| Antibody | Streptavidin Conjugated Alexa Fluor 488 | ThermoFisher Scientific | Cat#S11223 | (1:500) |
| Commercial assay, kit | iFX-enhancer | Invitrogen | Cat#I36933 | |
| Chemical compound, drug | Picric acid | Electron Microscopy Sciences | Cat#I9556 | |
| Chemical compound, drug | Glutaraldehyde | Sigma | Cat#G-7651 | |
| Chemical compound, drug | Sucrose | Sigma-Aldrich | Cat#S7903 | |
| Chemical compound, drug | Glucose | Sigma-Aldrich | Cat#G8270 | |
| Chemical compound, drug | Sodium chloride | Sigma-Aldrich | Cat#S9888 | |
| Chemical compound, drug | Potassium chloride | Sigma-Aldrich | Cat#P9333 | |
| Chemical compound, drug | Sodium phosphate monobasic dihydrate | Sigma-Aldrich | Cat#71505 | |
| Chemical compound, drug | Sodium bicarbonate | Sigma-Aldrich | Cat#S6297 | |
| Chemical compound, drug | Calcium chloride | Sigma-Aldrich | Cat#21115 | |
| Chemical compound, drug | Magnesium chloride | Sigma-Aldrich | Cat#68475 | |
| Chemical compound, drug | Cesium methanesulfonate | Acros Organics | CAS: 2550-61-0 | |
| Chemical compound, drug | HEPES | Sigma-Aldrich | Cat#H3375 | |
| Chemical compound, drug | EGTA | Sigma-Aldrich | Cat#E3889 | |
| Chemical compound, drug | Sodium phosphocreatine | Sigma-Aldrich | Cat#P7936 | |
| Chemical compound, drug | QX-314 (Lidocaine N-ethyl bromide) | Sigma-Aldrich | Cat#L5783 | |
| Chemical compound, drug | Magnesium-ATP | Sigma-Aldrich | Cat#A9187 | |
| Chemical compound, drug | Sodium-GTP | Sigma-Aldrich | Cat#G8877 | |

*Continued on next page*

*Continued*

| Reagent type | Reagent or resource | Source | Identifier | Additional information |
|---|---|---|---|---|
| Chemical compound, drug | Potassium gluconate | Sigma-Aldrich | Cat#G4500 | |
| Chemical compound, drug | DNQX | Tocris | Cat#0189; CAS: 2379-57-9 | |
| Chemical compound, drug | D-APV | Tocris | Cat#0106; CAS: 79055-68-8 | |
| Chemical compound, drug | Gabazine (SR 95531 hydrobromide) | Tocris | Cat#1262; CAS: 104104-50-9 | |
| Chemical compound, drug | Clozapine-*N*-oxide | Enzo Life Sciences | Cat#BML-NS105 | |
| Chemical compound, drug | Phosphate Buffered Saline (PBS) Tablets, 100 mL | VWR | Cat#E404-200TABS | |
| Chemical compound, drug | Triton X-100 | VWR | Cat#0694–1L | |
| Chemical compound, drug | Normal Donkey Serum | Jackson Immuno-research Labs Inc | Cat#017-000-121 | |
| Chemical compound, drug | Bovine Serum Albumin | Sigma-Aldrich | Cat#A9647-100G | |
| Commercial assay, kit | Prolong Gold antifade reagent | Invitrogen | Cat#P36934 | |
| Other | Nail Polish | Electron Microscope Sciences | Cat#72180 | |
| Chemical compound, drug | Biocytin | ThermoFisher | Cat#28022 | 0.2–0.5% |
| Other | Raw and analyzed data | This paper | Available on github and by request | https://github.com/emguthman/Manuscript-Codes (copy archived at https://github.com/elifesciences-publications/Manuscript-Codes) |
| Strain, strain background (*Mus musculus*) | Mouse: C57BL/6J | Jackson Lab | RRID: IMSR_JAX:000664 | All sexes used |
| Strain, strain background (*Mus musculus*) | Mouse: Sst$^{tm2.1(cre)Zjh}$/J | Jackson Lab | RRID: IMSR_JAX:013044 | All sexes used |
| Strain, strain background (*Mus musculus*) | Mouse: B6;129P2-Pvalb$^{tm1(cre)Arbr}$/J | Jackson Lab | RRID: IMSR_JAX:008069 | All sexes used |
| Strain, strain background (*Mus musculus*) | Mouse: B6;129S6-Gt(ROSA)26Sor$^{tm9(CAG-tdTomato)Hze}$/J | Jackson Lab | RRID: IMSR_JAX:007905 | All sexes used |
| Strain, strain background (*Mus musculus*) | Mouse: B6N.129-Gt(ROSA)26Sor$^{tm1(CAG-CHRM4*,-mCitrine)Ute}$/J | Jackson Lab | RRID: IMSR_JAX:026219 | All sexes used |

*Continued on next page*

*Continued*

| Reagent type | Reagent or resource | Source | Identifier | Additional information |
|---|---|---|---|---|
| Software, algorithm | pClamp 10.6 | Molecular Devices | RRID: SCR_011323 | |
| Software, algorithm | MATLAB_R2018a | Mathworks | RRID: SCR_001622 | |
| Software, algorithm | R | http://www.r-project.org/ | RRID: SCR_001905 | |
| Software, algorithm | Illustrator | Adobe | RRID: SCR_010279 | |
| Software, algorithm | Photoshop | Adobe | RRID: SCR_014199 | |
| Software, algorithm | SlideBook 6.0 | Intelligent Imaging Innovations (3i) | RRID: SCR_014300 | |
| Software, algorithm | Fiji | Fiji | RRID: SCR_002285 | |
| Software, algorithm | Grid/Collection Stitching | Fiji | RRID: SCR_016568 | |
| Software, algorithm | Neurolucida | MBF Biosciences | RRID: SCR_001775 | |

## Contact for reagent and resource sharing

Further information and requests for resources and reagents should be directed to and will be fulfilled by the lead contact, Molly M Huntsman (molly.huntsman@ucdenver.edu).

## Experimental model and subject details

All experiments were conducted in accordance with protocols approved by the Institutional Animal Care and Use Committee at the University of Colorado Anschutz Medical Campus. Slice electrophysiology experiments were conducted on mice aged postnatal days 35–70. Immunohistochemistry experiments were conducted on mice aged postnatal days 60–120. Experiments were conducted regardless of the observed external genitalia of the mice at weaning. To track estrous, vaginal swabs were collected from mice with vaginas that were used in experiments. The following mouse lines were used in the experiments: C57Bl/6J (Jackson Lab #000664), Sst-tdTomato, PV-tdTomato, Sst-hM4Di, PV-hM4Di, and wildtype littermates of the Sst-hM4Di and PV-hM4Di mice. Sst-tdTomato, PV-tdTomato, Sst-hM4Di, and PV-hM4Di mouse lines were generated by crosses of Sst-*ires*-Cre (Jackson Lab #013044) and PV-*ires*-Cre (Jackson Lab #008069) lines with either the Rosa-CAG-LSL-tdTomato-WPRE (*Ai9*; Jackson Lab #007905) or the R26-hM4Di/mCitrine (Jackson Lab #026219) lines. Please see Key Resources Table for more details on strain information.

## Method details

### Acute slice preparation for electrophysiology

Animals were first anesthetized with $CO_2$ and decapitated. Brains were quickly dissected and placed in an ice-cold, oxygenated (95% $O_2$-5% $CO_2$) sucrose-based slicing solution (in mM: sucrose, 45; glucose, 25; NaCl, 85; KCl, 2.5; $NaH_2PO_4$, 1.25; $NaHCO_3$, 25; $CaCl_2$, 0.5; $MgCl_2$, 7; osmolality, 290–300 mOsm/kg). 300–400 µm horizontal slices were obtained using a vibratome (Leica Biosystems, Buffalo Grove, IL, USA). Slices were incubated in oxygenated (95% $O_2$-5% $CO_2$) artificial cerebral spinal fluid (ACSF; in mM: glucose, 10; NaCl, 124; KCl, 2.5; $NaH_2PO_4$, 1.25; $NaHCO_3$, 25; $CaCl_2$, 2; $MgCl_2$, 2; osmolality 290–300 mOsm/kg) at 36 °C for at least 30 min. All reagents were purchased from Sigma-Aldrich (St. Louis, MO, USA).

### Electrophysiology

Slices were placed in a submerged slice chamber and perfused with ACSF heated to 32-37°C. Slices were visualized using a moving stage microscope (Scientifica: Uckfield, UK; Olympus: Tokyo, Japan)

equipped with 4× (0.10 NA) and 40× (0.80 NA) objectives, differential interference contrast (DIC) optics, infrared illumination, LED illumination (CoolLED, Andover, UK), a CoolSNAP EZ camera (Photometrics, Tuscon, AZ, USA), and Micro-Manager 1.4 (Open Imaging, San Francisco, CA, USA). Whole cell patch clamp recordings were made using borosilicate glass pipettes (2.5-5.0 MΩ; King Precision Glass, Claremont, CA, USA) filled with intracellular recording solution. For voltage clamp experiments on FFI a cesium methanesulfonate (CsMe) based intracellular solution was used (in mM: CsMe, 120; HEPES, 10; EGTA, 0.5; NaCl, 8; Na-phosphocreatine, 10; QX-314, 1; MgATP, 4; Na$_2$GTP, 0.4; pH to 7.3 with CsOH; osmolality adjusted to approximately 290 mOsm/kg). For all remaining voltage clamp experiments and for all current clamp experiments, a potassium gluconate based intracellular solution was used (in mM: potassium gluconate, 135; HEPES, 10; KCl, 20; EGTA, 0.1; MgATP, 2; Na$_2$GTP, 0.3; pH to 7.3 with KOH; osmolality adjusted to approximately 295 mOsm/kg). A subset of the potassium gluconate recordings were supplemented with 0.2-0.5 % biocytin to allow for post-hoc morphological analysis. Access resistance was monitored throughout the experiments and data were discarded if access resistance exceeded 25 MΩ or varied by more than ± 20%. No junction potential compensation was performed. Data were acquired with a Multiclamp 700B amplifier and were converted to a digital signal with the Digidata 1440 digitizer using pCLAMP 10.6 software (Molecular Devices, Sunnyvale, CA). Data were sampled at 10 kHz and lowpass filtered at 4 kHz. Offline, current data were filtered using a 3$^{rd}$ order Savistky-Golay filter with a ± 0.5 ms window after access resistance was assessed. Mean traces were created by first aligning all events by their point of maximal rise (postsynaptic currents) or by threshold (APs) and then obtaining the mean of all events; mean subthreshold EPSPs were not aligned prior to averaging.

## Cell-type identification
### Principal Neurons (PNs)
PNs were targeted based on their large, pyramidal-like soma. Recordings were terminated if the physiology of the neuron was inconsistent with BLA PNs (e.g. high membrane resistance, narrow AP halfwidth, large and fast spontaneous EPSCs).

### Interneurons (INs)
INs were targeted based on fluorescence in the Sst-tdTomato, PV-tdTomato, SST-hM4D$_i$, and PV-hM4D$_i$ mouse lines. A 470 nm LED was used to identify mCitrine[+] INs in SST-hM4D$_i$ and PV-hM4D$_i$ mouse lines, and a 535 nm LED was used to identify tdTomato[+] INs in the SST-tdTomato and PV-tdTomato mouse lines (CoolLED, Andover, UK).

## Pharmacology
DNQX, D-APV, and gbz were purchased from Tocris Biosciences (Bristol, UK) and CNO was purchased from Enzo Life Sciences (Farmingdale, NY). DNQX stock was made at 40 mM and diluted to a final concentration of 20 µM in ACSF; D-APV stock was made at 50 mM and diluted to a final concentration of 50 µM in ACSF; GBZ stock was made at 25 mM and diluted to a final concentration of 5 µM in ACSF; and, CNO stock was made at 10 mM and diluted to a final concentration of 10 µM in ACSF. All stocks were stored at −20°C and CNO was used within one month of making the stock solution.

## Electrophysiology experimental design
### FFI, voltage clamp
The LEC was stimulated using a bipolar stimulating electrode (FHC, Inc, Bowdoin, ME, USA). Evoked EPSCs ($V_{hold} = -70$ mV) and IPSCs ($V_{hold} = 0$ mV) were recorded from BLA PNs in response to LEC stimulation. To assess the effects of different drugs on the EPSCs and IPSCs, ACSF containing DNQX, D-APV, gbz, and/or CNO was perfused onto the slice for five minutes prior to and continuously during the experiment. Effects of DNQX/APV and gbz on EPSCs and IPSCs were recorded using in an unpaired design where some PNs were recorded under control conditions and in the presence of DNQX/APV and gbz (given sequentially with time for washout) whereas others were recorded under control conditions in the presence of DNQX/APV or gbz. Effects of CNO on EPSCs and IPSCs were examined with a paired design where all PNs were recorded in both control conditions and in the presence of CNO.

### FFI, current clamp

Membrane voltage of BLA PNs was recorded in response to 5 stimulations of the LEC at 20 Hz. $I_{hold}$ was adjusted such that $V_{rest}$ of the PNs was approximately $-60$ mV. To assess the role of $GABA_A$ receptor mediated inhibition on PN AP firing, gbz was perfused onto the slice for five minutes prior to and continuously during the experiment.

### Current injections

Membrane voltage of BLA neurons was recorded in current clamp in response to a series of square hyperpolarizing and depolarizing current injections. Prior to initiation of the series of current injections, $V_m$ of the BLA neurons was adjusted to approximately $-60$ mV. Each cell was subjected to two series of 600 ms square current injections: $-100$ pA to $+100$ pA at 10 pA intervals and $-200$ pA to $+400$ pA at 25 pA intervals. The data collected in these experiments were used to determine active and passive membrane properties of the neurons.

### Minimal stimulation

EPSCs ($V_{hold} = -70$ mV) were recorded in voltage clamp in BLA PNs, Sst$^+$ INs, and PV$^+$ INs in response to LEC stimulation. Stimulation intensity was adjusted such that LEC stimulation resulted in recorded EPSCs having a success rate of approximately 50% and an all-or-none amplitude response.

### Recruitment of BLA INs by LEC afferents

The median stimulation intensity necessary to observe putative uEPSCs in BLA PNs (273 μA × ms) was used as the empirically derived PN threshold stimulation intensity. Membrane voltage responses of BLA Sst$^+$ and PV$^+$ INs were recorded in current clamp in response to 5 stimulations of the LEC at 20 Hz at the empirically derived PN threshold stimulation intensity.

### Paired PN-IN recordings

Paired recordings were made between BLA PNs and nearby Sst$^+$ or PV$^+$ INs. We used a 2.5 nA, 2 ms current injection to drive a single AP in the PN ($I_{hold}$ adjusted such that $V_m \approx$ -60 mV. BLA PN APs were repeated at 0.25 Hz and the response of the IN ($V_{hold}$ = -70 mV) was recorded.

### Spontaneous EPSC recordings

sEPSCs were recorded for 5 min in INs ($V_{hold} = -70$ mV) with no drugs in the bath.

### Pharmacological effects of CNO on membrane potential

Membrane voltage of mCitrine$^+$ neurons in BLA was recorded in the presence of 20 μM DNQX, 50 μM D-APV, and 5 μM GBZ. To assess the effects of CNO on membrane potential, baseline $V_m$ was allowed to stabilize and was recorded for 3 min in the absence of CNO. Following recording of baseline $V_m$, ACSF containing 10 μM CNO was perfused onto the slice and $V_m$ was recorded for an additional 10 min. $V_m$ was separated into 30 s bins. $\Delta V_m$ was defined as the difference in $V_m$ between the mean $V_m$ during the 3 min of baseline recordings and the mean $V_m$ during the last 3 min of CNO application.

## Definitions of electrophysiological parameters

### Evoked EPSC/IPSC detection and amplitude

EPSCs ($V_{hold}$ = -70 mV) were defined as negatively deflecting postsynaptic events that exceeded the mean baseline current (500 ms before stimulation) by 6 × the median absolute deviation of the baseline current and that occurred within 20 ms of the end of the electrical stimulus artifact. IPSCs ($V_{hold}$ = 0 mV) were defined as positively deflecting postsynaptic events that exceeded the mean baseline current by 6 × the median absolute deviation of the baseline current and that occurred within 20 ms of the end of the electrical stimulus artifact. To ensure that the detected EPSCs/IPSCs were related to the stimulus, we subsampled 25% of the sweeps in the experiment (or 5 sweeps if the experiment consisted of <20 sweeps) and found the maximal peak negative (EPSC detection) or positive (IPSC detection) deflection from baseline in the 20 ms after the stimulus artifact in those sweeps. Then, we found the median peak time for those sweeps and repeated the analysis over all sweeps in the experiment with a detection threshold of 6 × the median absolute deviation of the baseline current and with a window set to ± 5 ms (EPSC detection) or ± 7.5 ms (IPSC detection) around the median

peak time. The amplitude of each EPSC and IPSC was defined as the difference between the peak amplitude of the detected EPSC or IPSC and the mean baseline current for that sweep. The EPSC or IPSC amplitude for each cell in an experiment was defined as the mean of the amplitudes recorded from that cell (EPSC/IPSC successes only; failures were not included in EPSC/IPSC amplitude calculation). Where EPSC failure amplitude is reported on a per sweep basis, it was defined as the maximal negative deflection from the current trace within $\pm$ 5 ms of the mean current peak time for that cell. If all sweeps in an experiment were EPSC/IPSC failures, the mean EPSC/IPSC amplitude was defined as the maximal negative (EPSC) or positive (IPSC) deflection in the mean current trace that occurred within 20 ms of the end of the electrical stimulus artifact.

## Success rate
Success rate was defined as $100\% \times \frac{n_{successful\ sweeps}}{n_{total\ sweeps}}$ where successful sweeps were defined as sweeps where an EPSC was detected.

## I/E balance
I/E balance was defined as the ratio of IPSC to EPSC amplitude recorded in the same neuron.

## EPSC/IPSC 20–80% risetime
20–80% risetime was defined as the time it took an EPSC or IPSC to reach 80% of its peak amplitude from 20% of its peak amplitude. 20–80% risetime was calculated for each sweep unless obscured by a spontaneous event. The risetime for each cell was defined as the mean of all risetimes recorded from that cell.

## EPSC/IPSC latency and jitter
Latency of EPSCs and IPSCs was defined as the time between the end of the electrical stimulation or the peak of the PN action potential (in paired recordings) and the point of 20% rise for an EPSC or IPSC as calculated for the 20–80% risetime. The EPSC or IPSC latency for each cell was defined as the mean of the latencies recorded from that cell. The EPSC or IPSC jitter for each cell was defined as the standard deviation of the latencies recorded from that cell.

## EPSC/IPSC $\tau_{Decay}$
EPSC $\tau_{Decay}$ was determined using a single exponential fit, $f(t) = Ae^{-t/\tau}$. IPSC $\tau_{Decay}$ was defined as the weighted time-constant of IPSC decay. Briefly, a double exponential fit, $f(t) = A_1 e^{-t/\tau_1} + A_2 e^{-t/\tau_2}$, was used to obtain the parameters to determine the weighted time-constant where $\tau_{weighted} = (\tau_1 A_1 + \tau_2 A_2)/(A_1 + A_2)$. $\tau_{Decay}$ was calculated using the mean EPSC or IPSC trace for a cell.

## uEPSC detection and amplitude
uEPSCs were defined as negative current deflections recorded in the IN that exceeded a detection threshold of 6x the median absolute deviation of the baseline current and occurred within 3ms of the PN AP peak during paired recordings. The amplitude of each uEPSC was defined as the difference between the peak amplitude of the detected uEPSC and the mean baseline current for that sweep. The uEPSC amplitude for each cell in an experiment was defined as the mean of the amplitudes recorded from that cell (successes only).

## sEPSC detection and amplitude
sEPSCs were detected by a combined template and threshold method. Briefly, a template was made by subsampling 10% of local negative peaks exceeding at least 5 $\times$ the median absolute deviation of a rolling baseline current (50ms prior to the peak). The template current was then truncated from its 20% rise point through the end of the decay time constant for the template current. Next, all local negative peaks exceeding 5 $\times$ the median absolute deviation of a rolling baseline current (50ms prior to the peak) were collected. The template current was then scaled to each individual putative sEPSC peak and each sEPSC peak was assigned a normalized charge integral relative to the template. Finally, a normalized charge integral cutoff was chosen to exclude obvious noise/non-

physiological events below a certain normalized charge integral. sEPSC amplitude was defined as the difference between the peak amplitude of each detected current and its corresponding baseline current. sEPSC for each cell was defined as the median peak amplitude for that cell.

## sEPSC frequency

sEPSC frequency for each sESPC was defined as the inverse of the interevent intervals of the sEPSCs. The frequency measure for each neuron was defined as the median of the sEPSC frequencies for that cell.

## Membrane resistance

Membrane resistance was defined as the slope of the best fit line of the I-V plot using the −100 pA to +100 pA (10 pA steps) series of current injections. Mean voltage response to each current injection step was defined as the difference between baseline mean membrane voltage (100 ms prior to current injection) and the mean membrane voltage during the 100 ms period from 50 ms after the start of the injection to 150 ms after the start of the current injection. This 100 ms window was chosen to allow for measurement of the change in $V_m$ after the membrane had charged and prior to any potential HCN channel activation. The I-V plot was constructed using all current steps below rheobase.

## Maximum firing rate

Maximum firing rate was defined as the inverse of the inter-spike interval (ISI) during the first 200 ms of the most depolarizing current injection step before attenuation of AP firing was observed. Max FR was calculated using the −200 pA to +400 pA (25 pA steps) series of current injections.

## AP threshold

AP threshold was defined as the voltage at which $\frac{dV}{dt}$ exceeded 20 V/s. AP threshold was calculated at the rheobase sweep of the -200 pA to +400 pA (25 pA steps) series of current injections.

## AP amplitude

Amplitude of the AP was defined as the voltage difference between the peak of the AP and its threshold potential. AP amplitude was calculated at the rheobase sweep of the −200 pA to +400 pA (25 pA steps) series of current injections.

## AP halfwidth

AP halfwidth was defined as the time between the half-amplitude point on the upslope of the AP waveform to the half-amplitude point on the downslope of the AP waveform. AP halfwidth was calculated at the rheobase sweep of the −200 pA to +400 pA (25 pA steps) series of current injections.

## After-hyperpolarization potential (AHP) magnitude

AHP magnitude was defined as the difference between the most hyperpolarized membrane voltage of the AHP (occurring within 100 ms after AP threshold) and AP threshold. AHP magnitude and latency data were calculated at the rheobase sweep of the −200 pA to +400 pA (25 pA steps) series of current injections. ΔAHP data were calculated at the rheobase + 50 pA sweep of the −200 pA to +400 pA (25 pA steps) series of current injections.

## AHP latency

AHP latency was defined as the time from AP threshold and the peak of the AHP.

## ΔAHP

ΔAHP was defined as the difference between the first and last AHP ($\Delta AHP = AHP_{last} - AHP_{first}$).

## AP phase plot

The AP phase plot was obtained by plotting the rate of change of the mean AP for each cell from the rheobase sweep of the −200 pA to +400 pA (25 pA steps) series of current injections as a function of the corresponding membrane voltage.

## Latency to first AP

AP latency was defined as the time from the initiation of the current injection to the peak of the first AP. AP latency was calculated at the rheobase sweep of the −200 pA to +400 pA (25 pA steps) series of current injections.

## Firing rate adaptation ratio (FR adaptation)

Firing rate adaptation was defined as the ratio of the first and the average of the last two ISIs, such that $Firing\ rate\ adaptation = \frac{ISI_{first}}{meanISI_{last\ 2ISI}}$. Firing rate adaptation was calculated at the rheobase +50 pA sweep of the -200 pA to +400 pA (25 pA steps) series of current injections.

## AP broadening

AP broadening was defined as the ratio of the AP halfwidths of the first two APs ($Broadening = \frac{halfwidth_{second}}{halfwidth_{first}}$). AP broadening was calculated at the rheobase +50 pA sweep of the -200 pA to +400 pA (25 pA steps) series of current injections.

## AP amplitude adaptation

AP amplitude adaptation was defined as the ratio of the AP amplitude of the average of the last three APs and the first AP, such that $AP\ amplitude\ adaptation = \frac{meanAmplitude_{last\ 3APs}}{Amplitude_{first}}$. AP amplitude adaptation was calculated at the rheobase +50 pA sweep of the -200 pA to +400 pA (25 pA steps) series of current injections.

## Membrane decay τ

Membrane decay τ was determined by using a single exponential fit, $f(t) = Ae^{-t/\tau}$, to fit the change in $V_m$ induced by a −100 pA sweep in the −100 pA to +100 pA (25 pA steps) series of current injections.

## Hyperpolarization-induced sag

Hyperpolarization-induced sag was calculated using the equation, $\frac{V_{min}-V_{ss}}{V_{min}-V_{bl}} \times 100\%$, where $V_{min}$ was defined as the most hyperpolarized membrane voltage during the current injection, $V_{ss}$ was defined as the mean steady-state membrane voltage (last 200 ms of the current injection), and $V_{bl}$ was defined as the mean baseline membrane voltage (100 ms prior to current injection). Hyperpolarization-induced sag was measured from the -200 pA current injection.

## Rebound spikes

Rebound spikes were defined as the number of APs in the 500 ms following the −200 pA current injection.

## APs per stimulus

The number of APs per stimulus was defined as the number of APs occurring within 50 ms of the stimulus.

## $V_{rest}$

$V_{rest}$ was defined as $V_m$ ($I_{hold}$ = 0 pA) during a 500 ms baseline prior to LEC stimulation during the experiments testing the recruitment of BLA INs by LEC afferents.

## Subthreshold EPSP amplitude

Subthreshold EPSP amplitude was defined as the maximal, non-stimulus artifact, voltage deflection within 40 ms after LEC stimulation.

## Immunohistochemistry
### Biocytin filled neurons

To perform immunostaining of biocytin filled neurons, slices containing biocytin filled neurons were fixed in 4% PFA overnight at 4°C. After fixation, slices were transferred to PBS. Biocytin filled INs (n = 8 $Sst^+$ Group 1; 3 $Sst^+$ Group 2; 6 $PV^+$) were blocked (1X PBS, 0.3% triton, 5% BSA, 5% Normal Donkey Serum) for 4 hours before 24-hour incubation with streptavidin conjugated Alexa Fluor 488 (1:500, ThermoFisher Scientific, Waltham, MA, USA) at 4°C. Slices were mounted with Prolong Gold and sealed for long-term storage. Slices were imaged using an Axio Observer microscope (Carl Zeiss, Okerkochen, Germany); equipped with a CSU-X1 spinning disc unit (Yokogawa, Musashino, Tokyo, Japan); 488 nm/40 mW laser; Plan-NeoFluar 40X (0.75 NA) air objective lens; and Evolve 512 EM-CCD camera (Photometrics, Tucson, AZ, USA). SlideBook 6.0 software (3i, Denver, CO, USA) enabled instrument control and data acquisition. Images were acquired in sections by following branched points from the cell soma. Images were stitched in Fiji software using Grid/Collection stitching (*Preibisch et al., 2009*) with an unknown position type.

### Histological validation of $PV^+$ and $Sst^+$ IN identity

To perform the PV and Sst immunostaining, mice (n = 3 Sst-tdTomato mice, 3 PV-tdTomato mice) were sacrificed and transcardially perfused with ice cold 4% PFA (with 1.5% picric acid and 0.05% glutaraldehyde) followed by 30% sucrose protection. After the brain sank, coronal BLA slices of 30 μm thickness were obtained. Before application of blocking solution, slices were incubated for 30 minutes at room temperature with iFX-enhancer (Invitrogen, ThermoFisher Scientific, Waltham, MA, USA). After blocking, the slices were incubated with either rat anti-Sst antibody (1:100, MAB354, Millipore, Burlington, MA, USA) or guinea pig anti-PV antibody (1:500, 195004, Synaptic Systems, Göttingen, Germany) for at least 48 hours at 4°C. Then, a secondary antibody of either donkey anti-rat or donkey anti-guinea pig Alexa Fluor 488 (1:500, Jackson ImmunoResearch) was applied over-night at 4°C. The slices were imaged using a confocal laser scanning microscope (TCS SP5II, Leica Application Suite, Leica Biosystems, Buffalo Grove, IL, USA) with 10 × 0.40 NA and 20 × 0.70 NA dry objectives to determine the neuron identity.

## Morphological analysis
### Analysis of somatic and dendritic morphology

To determine morphological characteristics of biocytin filled neurons, stitched images were imported to Neurolucida (MBF Bioscience, Williston, VT, USA) to perform tracing. All analysis including sholl analysis (50 μM rings), dendrite branching, etc. was performed from traces using Neurolucida Explorer (MBF Bioscience).

## Quantification and statistical analysis
### Statistical analyses

All data analysis (except decision tree and random forest analyses) were performed offline using custom written MATLAB code. Normality of the data was assessed using the Anderson-Darling test. For assessment of whether a single group differed from a normal distribution centered around zero, a one-sample t-test was used. For a test between two groups, a paired or unpaired t-test was used where appropriate. For tests between two groups of non-normal data, a Mann-Whitney U or Wilcoxon signed-rank test was used where appropriate. For tests between three or more groups of normal data with one independent variable, a one-way ANOVA was used with Tukey's post-hoc test to examine differences between groups. A Kruskal-Wallis test was used to examine differences between three or more groups of non-normal data with one independent variable. A Mann-Whitney U test was used as a post-hoc test following a significant result in a Kruskal-Wallis test and was corrected for multiple comparisons using the FDR method (*Curran-Everett, 2000*). The critical significance value was set to $\alpha = 0.05$ or was set to a FDR-corrected value ($\alpha_{FDR}$) for multiple comparisons. All statistical tests were two-tailed. Unless otherwise stated, experimental numbers are reported as

n = x, y where x is the number of neurons and y is the number of mice. Statistical parameters are reported in the Results section and figure legends display p values and sample sizes.

## Unsupervised cluster analysis

Unsupervised cluster analysis using Ward's method (*Ward, 1963*) was used to classify Sst[+] INs. Briefly, this method involves plotting each neuron in multidimensional space where each dimension corresponds to a given parameter. For our data, we plotted each Sst[+] IN in 15-dimensional space (where each dimension corresponds to a z-score transformation of one of the 15 membrane properties obtained from the current injection experiments; values were z-score transformed so that parameters with large values, for example membrane resistance, would not influence the cluster analysis more than those with small values, for example halfwidth). From here, the analysis proceeded along $n - one$ stages where $n$ is the number of Sst[+] INs. At stage $n = 1$, the two closest cells in the 15-dimensional space are grouped together. At subsequent stages, the closest cells are grouped together until only one group of all objects remains. We determined the final number of clusters by using the Thorndike procedure (*Thorndike, 1953*) where large distances between group centroids at one cluster stage relative to other stages are indicative of significant differences between groups (see *Figure 3A*, inset).

## Decision tree analysis

Recursive partitioning analyses (*Breiman et al., 1984*) were conducted in R (1.01.136). The R package rpart (*Therneau and Atkinson, 1984*) was used for recursive partitioning for classification. Decision trees were plotted using rpart.plot package, and random seeds were set using the rattle package. Predicted classes were determined from the unsupervised cluster analysis and were used to determine important discriminating parameters. The model was internally cross validated with a nested set of subtrees and the final model was selected as the subtree with the least misclassification error. Individual decision trees can suffer from overfitting. Therefore, the input parameters of the model were independently cross validated by bootstrapping 500 subsamples (without replacement and using random seeds). Each tree was pruned by choosing the complexity parameter with the lowest cross validated error. The mean correct prediction of the test set classifications (n = 20% of sample) by the pruned tree generated by modeling the training sets (n = 80% of sample) was 82.9%.

## Random forest analysis

Supervised classification random forest was employed to reduce potential model overfitting. Random forest classification was conducted using the randomForest package in R (*Liaw and Wiener, 2002*). Random forest classification employs random subsampling of both parameters and bootstrapped subsamples of the dataset (with replacement). 10,000 decision trees were generated, and the modal tree was used as the final classification model. The out of bag estimate of the error rate of the model was 11.43%. An additional cross validation step was conducted by bootstrapping 80% of the initial sample to create the random forest model and tested against a subsample (n = 20% of sample; bootstrapped without replacement). The mean accuracy after 20 random forest runs was 90.7%. Gini impurity was calculated to determine parameter importance. Visualization of the frequency of individual samples falling within the same node across one run of the classification algorithm was obtained by calculating a proximity matrix for the samples and plotting it against the first two principle components.

## Data display

Data visualizations were created in MATLAB and Adobe Illustrator. After analysis was completed, Neurolucida traces of soma and dendrites were thickened by 7 pixels (*Figure 3*) or 3-7 pixels (*Figure 3—figure supplement 2*) in Adobe Photoshop to improve visibility in figures. Normal data are presented as the mean ± s.e.m. Non-normal data are presented as the median with error bars extending along the interquartile range.

## Acknowledgements

We would like to thank Nicole Arevalo and the University of Colorado Anschutz Medical Campus Breeding Core for their assistance with animal care. We would also like to thank members of the Huntsman and Restrepo labs and Alicia M Purkey for discussions on the data. This work was supported by US National Institutes of Health grants (R01 DC000566 to DR, R01 NS095311 to MMH, T32 NS099042 to EMG, and T32 GM763540 to JDG) and US National Science Foundation Graduate Research Fellowship (DGE-1553798 to EMG).

## Additional information

### Funding

| Funder | Grant reference number | Author |
| --- | --- | --- |
| National Institutes of Health | R01 NS095311 | Molly M Huntsman |
| National Science Foundation | DGE-1553798 | E Mae Guthman |
| National Institutes of Health | R01 DC000566 | Diego Restrepo |
| National Institutes of Health | T32 NS099042 | E Mae Guthman |
| National Institutes of Health | T32 GM763540 | Joshua D Garcia |

The funders had no role in study design, data collection and interpretation, or the decision to submit the work for publication.

### Author contributions

E Mae Guthman, Conceptualization, Data curation, Formal analysis, Funding acquisition, Validation, Investigation, Visualization, Methodology; Joshua D Garcia, Ming Ma, Data curation, Methodology; Philip Chu, Data curation, Software, Formal analysis, Methodology; Serapio M Baca, Software; Katharine R Smith, Data curation, Formal analysis, Supervision; Diego Restrepo, Conceptualization, Supervision, Investigation; Molly M Huntsman, Conceptualization, Resources, Supervision, Funding acquisition, Investigation, Methodology

### Author ORCIDs

E Mae Guthman (ORCID) https://orcid.org/0000-0002-2190-7520
Diego Restrepo (ORCID) https://orcid.org/0000-0002-4972-446X
Molly M Huntsman (ORCID) https://orcid.org/0000-0002-5954-0023

### Ethics

Animal experimentation: This study was performed in strict accordance with the recommendations in the Guide for the Care and Use of Laboratory Animals of the National Institutes of Health. All of the animals were handled according to approved institutional animal care and use committee (IACUC) protocols (#00039) of the University of Colorado Denver | Anschutz Medical Campus.

### Decision letter and Author response

Decision letter https://doi.org/10.7554/eLife.50601.sa1
Author response https://doi.org/10.7554/eLife.50601.sa2

## Additional files

### Supplementary files

• Transparent reporting form

## Data availability

All data generated or analysed during this study are included in the manuscript and supporting files. Code is available on GitHub (https://github.com/emguthman/Manuscript-Codes, copy archived at https://github.com/elifesciences-publications/Manuscript-Codes).

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
