## [Decision Letter]

**Acceptance summary:**

This well written manuscript simplifies a very complex topic and clearly presents complicated electrophysiological studies. The study is conceptually sound and carefully conducted. This manuscript sheds light on an important, yet poorly understood, circuit impinging on the basolateral amygdala (BLA). This is a complex circuit and elucidating the functional dynamics of the local and extended circuitry of the BLA is challenging. This elegant electrophysiological study is a tour de force attempting to disentangle the function of the lateral entorhinal cortex (LEC) to BLA circuit that is implicated in olfactory learning. The data describe a novel fast-spiking SST^+^ interneuron in the BLA which is specifically recruited by LEC input and a suggests a unique mechanism that might gate plasticity in the BLA.

**Decision letter after peer review:**

Thank you for submitting your article "Cell type specific control of basolateral amygdala plasticity via entorhinal cortex-1 driven feedforward inhibition" for consideration by *eLife*. Your article has been reviewed by three peer reviewers, and the evaluation has been overseen by a Reviewing Editor and Laura Colgin as the Senior Editor. The following individual involved in review of your submission has agreed to reveal their identity: Jamie Maguire (Reviewer #2).

The reviewers have discussed the reviews with one another and the Reviewing Editor has drafted this decision to help you prepare a revised submission.

Essential revisions:

Overall, while the comprehensive approach is impressive and carefully teases out the circuits recruited by LEC input, the experiments on plasticity are underpowered and seem to rely on novel, i.e. non-standard, definitions of plasticity that might not be fully justified. The paper would be much stronger with additional experiments that would adequately power the plasticity results. However, given the extensive and important findings of the paper, the reviewers felt that even without the results on plasticity the paper would stand on its own. Accordingly, elimination of these results along with corresponding changes in title, Abstract, etc, would also be acceptable.

Detailed critique:

1) There are far too few cells in the analysis or the role of specific interneurons subtypes in plasticity induction. The investigators divide cells into "no plasticity" and LTP or LTD neurons, but there is no clear rationale for how those groups are created or why results are so heterogeneous. With such varied outcomes (and given the ambiguity in where exactly the recordings were located), it is important to increase the size of the dataset to determine whether there is a specific trend with inhibition blocked. It was unclear if the CNO application experiments were done blind – at least a portion of these should be carried out without investigator knowledge of either CNO presence or animal genotype. Also, Figure 1H-I would be better off in a separate figure or put together with Figure 6.

2) In neocortical areas, some SST neurons also express PV (see Hu, Cavendish, Agmon 2013 and others). Is this the case in the BLA? Are the authors suggesting that the FS-SST neurons are a novel subset of SST, or are PV cells that show off-target transgene expression? There are a number of studies that have investigated FS SST neurons; these studies should better cited. Can the authors stain the tissue with PV to see whether some SST neurons also show PV expression?

3) It is of concern that these data are incongruent with a prior study (Lucas et al., 2016). The authors suggest that they cannot accurately detect the boundaries between the BLA and LA in their horizontal prep. However, they indicate that their study was confined to the BLA throughout the paper. How was it verified? To better understand where they were recording from in their novel slice prep, they should target some injections to the BLA and then prepare their slice prep, to figure out their recording location. In addition, they should show a photo of the prep in their first figure, and indicate some anatomical landmarks that can be used by others in the field to take advantage of this slice preparation and replicate the results.

4) The Abstract talks about olfactory learning, but there is nothing in the paper that is related to this. Since it is rather speculative, it should be taken out.

5) The terminology and methodology around bidirectional plasticity should be improved and rationalized. If the authors are correct, then it is not so much bidirectional as heterogeneous, with some cells experiencing LTD, others LTP and yet others little effect at all. I think a very rigorous statistical approach is required to clearly demonstrate the existence of these distinct subpopulations over the more parsimonious explanation than on average there is little plasticity in this pathway. The experiments appear to be underpowered to really make this point.

---

## [Author Response]

Essential revisions:Overall, while the comprehensive approach is impressive and carefully teases out the circuits recruited by LEC input, the experiments on plasticity are underpowered and seem to rely on novel, i.e. non-standard, definitions of plasticity that might not be fully justified. The paper would be much stronger with additional experiments that would adequately power the plasticity results. However, given the extensive and important findings of the paper, the reviewers felt that even without the results on plasticity the paper would stand on its own. Accordingly, elimination of these results along with corresponding changes in title, Abstract, etc, would also be acceptable.

We would like to thank the reviewers for their critiques and suggested revisions to our manuscript. We agree that additional experiments would better power the plasticity results. We have made the requested revisions and removed these results with corresponding changes to the rest of the paper.

Detailed critique:1) There are far too few cells in the analysis or the role of specific interneurons subtypes in plasticity induction. The investigators divide cells into "no plasticity" and LTP or LTD neurons, but there is no clear rationale for how those groups are created or why results are so heterogeneous. With such varied outcomes (and given the ambiguity in where exactly the recordings were located), it is important to increase the size of the dataset to determine whether there is a specific trend with inhibition blocked. It was unclear if the CNO application experiments were done blind – at least a portion of these should be carried out without investigator knowledge of either CNO presence or animal genotype. Also, Figure 1H-I would be better off in a separate figure or put together with Figure 6.

We thank the reviewers for this critique. As mentioned above, these data (old Figure 1H-I and old Figure 6) have been removed from the current manuscript.

2) In neocortical areas, some SST neurons also express PV (see Hu, Cavendish, Agmon 2013 and others). Is this the case in the BLA? Are the authors suggesting that the FS-SST neurons are a novel subset of SST, or are PV cells that show off-target transgene expression? There are a number of studies that have investigated FS SST neurons; these studies should better cited. Can the authors stain the tissue with PV to see whether some SST neurons also show PV expression?

Yes, there is some, albeit small (~16%), overlap between PV and Sst expression in the BLA. We stained tissue for PV expression in the Sst-tdTomato and for Sst expression in the PV-tdTomato mouse lines. These data were presented as Figure 2—figure supplement 2 and show 13.28 ± 1.11% overlap between PV and tdTomato expression in the Sst-tdTomato mouse line and 18.34 ± 2.24% overlap between Sst and tdTomato expression in the PV-tdTomato mouse line. These data were discussed the original submission and in the first paragraph of the subsection “Cell type specificity of the LEC→BLA circuit” in the new submission.

We propose that the FS Sst INs are a novel subset of Sst INs rather than a population of PV cells with off-target transgene expression. Our major arguments for this are the following:

1) FS Sst INs accounted for 57% of all Sst INs (Figure 2—source data 2). The PV/Sst overlap of ~16% in our transgenic lines (Figure 2—figure supplement 2) cannot account for the fraction of FS Sst INs.

2) If these FS Sst neurons were actually PV neurons with off-target cre expression in the Sst-cre mouse lines, we would expect to see a decrease in FFI when we inactivate PV INs. However, in Figure 5, we show that inactivation of Sst INs but not PV INs led to a decrease in FFI.

We have updated the Discussion to better cite other studies that have investigated the FS Sst INs. This discussion can be found in the third paragraph of the Discussion.

3) It is of concern that these data are incongruent with a prior study (Lucas et al., 2016). The authors suggest that they cannot accurately detect the boundaries between the BLA and LA in their horizontal prep. However, they indicate that their study was confined to the BLA throughout the paper. How was it verified? To better understand where they were recording from in their novel slice prep, they should target some injections to the BLA and then prepare their slice prep, to figure out their recording location. In addition, they should show a photo of the prep in their first figure, and indicate some anatomical landmarks that can be used by others in the field to take advantage of this slice preparation and replicate the results.

We think there has been a miscommunication on our end. The LA and BA are the two subnuclei that make up the BLA as a whole. This miscommunication likely arises from older literature calling the basal nucleus the basolateral nucleus of the basolateral amygdala. In the literature, the basal/basolateral nucleus of the basolateral amygdala has been abbreviated as BA when the basal amygdala nomenclature is used or BL when it is called the basolateral nucleus. Some authors have further subdivided the BA/BL into anterior and posterior regions, naming them BLA and BLP (not to be confused with the basolateral amygdala as a whole, this BLA is the anterior basolateral nucleus of the basolateral amygdala, a subnucleus of a subnucleus of the totality of the basolateral amygdala). For an example of this nomenclature, please see Author response image 1, an image from Paxinos and Franklin’s the Mouse Brain in Stereotaxic Coordinates, 4^th^ ed. where the BA is called the BL with subsequent subdivision into BLA and BLP.

This nomenclature is confusing and can lead to a misunderstanding of which regions are being recorded from. For that reason, we referred to the two major subnuclei as LA and BA and the whole basolateral amygdala as BLA as this has become the standard nomenclature in the field. For examples of the LA/BA nomenclature (including in use in the horizontal slice) please see the above reference paper by Lucas and colleagues, the below paper by Andrasi and colleagues, and a recent review from Janak and Tye (Andrasi et al., 2017; Lucas et al., 2016; Janak and Tye, 2015).

With regards to our experimental preparation, we cannot accurately detect the boundaries between LA and BA in our horizontal prep without posthoc staining for VAChT, but we can unambiguously detect the boundaries of the BLA as a whole. (The only way to accurately detect LA and BA boundaries in the horizontal slice is with posthoc VAChT staining: Andrasi et al., 2017).

Similar to what is found in the coronal slice, the external capsule branches in two and creates medial and lateral boundaries around the BLA proper. In Author response image 1, we have included images of our slice prep that shows the BLA in the horizontal slice surrounded by these branches of the external capsule. The anatomical landmarks we used to define the BLA were taken from Paxinos and Franklin’s the Mouse Brain in Stereotaxic Coordinates, 4^th^ ed. A page from Paxinos showing the horizontal section of the mouse brain containing both the lateral and basal nuclei of the BLA is presented under the representative images of our recording and stimulation preparation to show that our recording sites correspond to the anatomical definitions of the BLA in the horizontal section.

All our recordings conform to the anatomically defined boundaries of the BLA. As should be clear from the slice images, no anatomical landmarks distinguish which subnuclei we recorded from, but we state with confidence that every recording was conducted within the BLA as a whole structure.

4) The Abstract talks about olfactory learning, but there is nothing in the paper that is related to this. Since it is rather speculative, it should be taken out.

We agree with the reviewers and have taken out our mention of olfactory learning in the Abstract.

5) The terminology and methodology around bidirectional plasticity should be improved and rationalized. If the authors are correct, then it is not so much bidirectional as heterogeneous, with some cells experiencing LTD, others LTP and yet others little effect at all. I think a very rigorous statistical approach is required to clearly demonstrate the existence of these distinct subpopulations over the more parsimonious explanation than on average there is little plasticity in this pathway. The experiments appear to be underpowered to really make this point.

We agree with the reviewers that the plasticity experiments could be improved. As mentioned above, at the reviewers’ suggestion, we have removed these experiments from the manuscript.